



# A method for assessment of the general circulation model quality using K-means clustering algorithm

Urmas Raudsepp[1], Ilja Maljutenko[1]

5   [1]Department of Marine Systems, Tallinn University of Technology, Tallinn, 19086, Estonia

*Correspondence to*: Urmas Raudsepp (urmas.raudsepp@taltech.ee)





**Abstract.** The model's ability to reproduce the state of the simulated object or particular feature or phenomenon is always a subject of discussion. Multidimensional model quality assessment is usually customized to the specific focus of the study and often to a limited number of locations. In this paper, we propose a method that provides information on the accuracy of the model in general, while all dimensional information for posterior analysis of the specific tasks is retained. The main goal of the method is to perform clustering of the multivariate model errors. The clustering is done using the K-means algorithm of unsupervised machine learning. In addition, the potential application of the K-means clustering of model errors for learning and predicting is shown. The method is tested on the 40-year simulation results of the general circulation model of the Baltic Sea. The model results are evaluated with the measurement data of temperature and salinity from more than one million casts by forming a two-dimensional error space and performing a clustering procedure in it. The optimal number of clusters that consist of four clusters was determined using the Elbow cluster selection criteria and based on the analysis of the different number of error clusters. In this particular model, the error cluster of good quality of the model with a bias of 0.4 °C (std=0.8 °C) for temperature and 0.6 g kg$^{-1}$ (std=0.7 g kg$^{-1}$) for salinity made up 57% of all comparison data pairs. The prediction of centroids from a limited number of randomly selected data showed that the obtained centroids gained a stability of at least 100 000 error pairs in the learning dataset.

## 1 Introduction

Ocean general circulation models are valuable tools for hindcasting and forecasting ocean state. The values of the simulated fields depend on the quality of the modeling products. Assessment of model quality is a basic step that is taken before the model results are used for evaluation of the ocean state or used for other specific purposes. For instance, product quality assessment is routinely done for all products of the Model Forecast Centers within Copernicus Marine Environment Monitoring Service (CMEMS, 2016) and National Oceanic and Atmospheric Administration (NOAA, https://www.esrl.noaa.gov/fiqas/, https://sats.nws.noaa.gov/~verification/; https://www.ncdc.noaa.gov/sotc/global/202101).

Ocean general circulation model output consists of a set of variables in space and time, i.e., 4-dimensional fields. The classical approach is that statistical metrics are calculated independently for each variable used for validation. In addition, the dimensionality of the output is frequently reduced by either doing certain averaging and/or selecting one or two dimensional subsets. Common statistical metrics for a single prognostic variable (e.g., bias, root mean square difference, correlation coefficient, standard deviations) are used to assess the model skills (Murphy et al., 1989; Murphy, 1995; Węglarczyk, 1998; Jolliff et al., 2009; Dybowski et al., 2019). Taylor diagrams (Taylor, 2001) or target diagrams (Joliff et al., 2009) are usually implemented for compact visualisation of the model performance statistics. Stow et al. (2009) studied 149 papers based on numerical modeling. They found that the majority (68%) of the model validation works were based on visual comparison and comparing simple statistics such as bias and variance, 9% of the works calculated the correlation coefficient and roughly 11% of the works implemented various cost-function techniques (e.g., Holt et al. 2005; Eilola et al., 2009). Even if all available data with sufficient spatio-temporal coverage is used for multivariate comparison, the end result is a single metric





or limited set of metrics that, indeed, characterize the model general quality. The shortcoming of this approach is that 4-dimensional information embedded in the huge dataset used for the validation will be lost.

Temperature and salinity are widely used state variables for the assessment of the accuracy of general circulation models. Ideally, researchers like to know the model accuracy for the whole model domain and time period considered. The amount of observational data has increased tremendously over the past decades. Temperature and salinity are usually measured

simultaneously and form a major share of the data in the databases.

We suggest a new method that takes advantage of a large set of all available data and belongs to the category of multivariate comparison. The method is not limited to the set of two variables. The only requirement is that all variables should be simultaneously measured. Preprocessing can be done to make data simultaneous, i.e., averaging over some space and time. The method is based on the machine learning K-means clustering algorithm (Jain, 2010).

he intuitive prerequisite for using any clustering approach is that the dataset should have a natural cluster structure (Jain, 2010). A prior knowledge about the model accuracy and distribution of model errors in space and time is usually missing. If there is a large number of data for comparison, then distribution of the model errors might not show visually identified clusters. If more than two variables are used for the model quality assessment, then visualisation of the errors for the identification of the clusters becomes more complicated.

In this study, we will show that implementing the K-means clustering algorithm for the analysis of model temperature and salinity errors provides meaningful information about the model accuracy. Clustering procedure using the K-means algorithm includes quantitative metrics for general assessment of the model performance. Posterior analysis of error clusters is an essential part of the proposed method and enables us to understand model data misfit and to explain the errors in relation to the dynamic features of the natural water basin under consideration.

The proposed clustering methods can also be used in learning-predicting sequence. The latter is important in the operational use of the model. The learning period consists of the model run for a certain period and error clustering. The learning period is for determining the number of clusters and the coordinates of the centroids. Based on the learning period error clustering, we can presume that a similar error distribution is valid for the forward model simulation results. During the predicting period, new available errors are added to the clusters. The coordinates of the centroids and other metrics are updated. The

value of this process lies in the fact that exploitation of the model simulation results can start before new validation is completed. In this study, we implement the learning-predicting sequence in the form of clustering stability tests.

We apply proposed K-means clustering methods for the assessment of model quality of the circulation model of the Baltic Sea. The model has been used for the analysis of long-term water circulation in the Gulf of Finland (Maljutenko and Raudsepp, 2019). Conventional model validation with station measurements of temperature and salinity is presented in

Maljutenko and Raudsepp (2014, 2019).



## 2 Materials and Methods

### 2.1 The Baltic Sea

The Baltic Sea (Fig. 1a) is a wide non-tidal estuary-type marginal sea with a longitudinal salinity between 0 and 20 g kg$^{-1}$ (Leppäranta and Myrberg, 2009; Omstedt et al. 2014). The longitudinal salinity gradient is maintained by saline water

inflows from the North Sea through Danish straits and freshwater input by rivers. Large volumes of saline water are transported to the Baltic Sea by the Major Baltic Inflows (MBI) that occur seldom (Mohrholz, 2018). The other smaller inflows occur almost every winter (Mohrholz, 2018; Raudsepp et al., 2018). Inflowing saline water spreads downstream into the Baltic Sea along the cascade of deep basins: the Bornholm Basin, Gdansk Basin and the Eastern Gotland Basin. The saline water of the Gotland basin is pushed into the western Gotland Basin and the Gulf of Finland. During the MBIs, dense

inflow water spreads along the bottom while other large volume inflows renew the halocline layer of the Baltic Sea. The permanent halocline in the Baltic Sea is at a depth of 60-80 m (Väli et al., 2013). The Gulf of Bothnia and the Gulf of Riga do not have a permanent halocline (Raudsepp, 2001). The Gulf of Finland has a very dynamic halocline due to intensive estuarine circulation (Maljutenko and Raudsepp, 2019), occasional stratification collapses due to reverse estuarine circulation (Elken et al., 2014; 2003) and winter mixing. Seasonal thermocline at a depth range of 10-30 meters starts to

develop in spring, reaches its maximum strength in summer and erodes in autumn. In the gulf-type regions of freshwater influence, like the Gulf of Finland (Maljutenko and Raudsepp, 2019) and the Gulf of Riga (Soosaar et al., 2014), seasonal thermocline coincides with seasonal halocline in spring and summer. During maximum river runoff in spring, the river bulge affects the salinity distribution in the coastal sea (Soosaar et al., 2016; Maljutenko and Raudsepp, 2019).

Salinity fronts are formed in the straits that connect different sub-basins of the Baltic Sea: between Kattegat and

southwestern Baltic Sea, the Gulf of Riga and Baltic Proper, the Gulf of Bothnia and Baltic Proper. The Danish straits and Kattegat are situated in a region with a very dynamic and strong front that separates the brackish Baltic sea water and the saline North Sea water (Nielsen, 2005). The Baltic Sea water of low salinity is transported towards the North Sea in summer, but saline water of the North Sea inflows to the Baltic Sea in winter (Mohrholz, 2018). A dynamic front is present in the transition area between the northeastern Baltic Proper and the Gulf of Finland, although that is a wide and deep area.

The Baltic Sea is seasonally ice-covered. Inter-annually variable and dynamic ice coverage (Raudsepp et al., 2020) has considerable effect on the evolution of the thermohaline fields in the Baltic Sea.





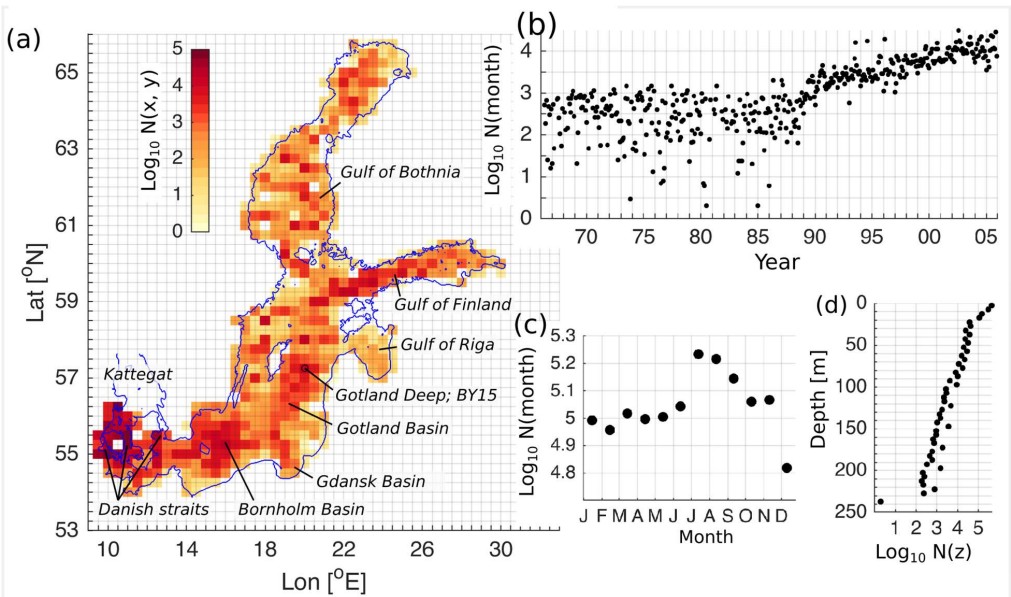

**Figure 1: Spatial (a), temporal (b), seasonal (c) and vertical (d) distribution of the number of measurements in the dataset. The horizontal bins have a resolution of 25x25 km (a), temporal and seasonal bins have monthly resolution (b,c), and vertical bins have a resolution of 5 m (d).**

## 2.2 Model simulation

The General Estuarine Transport Model (GETM; Burchard and Bolding, 2002) is a numerical 3D circulation model initially developed for coastal and estuarine applications (Gräwe et al., 2013; Holtermann et al., 2014). The hindcast simulation of the general circulation of the Baltic Sea was carried out for the period of 1966–2006 (Maljutenko and Raudsepp, 2019; 2014). Model open boundary was located in Kattegat where sea level elevation, temperature and salinity are prescribed. The model horizontal resolution was set to one nautical mile, which was consistent with the horizontal resolution of the digital bathymetry of the Baltic Sea (Seifert and Kayser, 1995). Vertically, 40 bottom-following adaptive layers were used, which resulted in a vertical resolution of less than 5 m.

The initial conditions of salinity and temperature were compiled using observation data from the Baltic Environmental Database (BED; http://nest.su.se/bed) (Gustafsson and Medina, 2011; Wulff et al., 2013). Atmospheric forcing was prepared from the BaltAn65+ reanalysis dataset (Luhamaa et al., 2011). The heat fluxes are parameterized using bulk formulation (Kondo, 1975). Monthly river runoff data from the 37 largest rivers from the E-HYPE hydrology model (Donnelly et al., 2016) were used. We have stored daily mean values of temperature and salinity and used them for the analysis.





### 2.3 Dataset

We use salinity and temperature measurements for the Baltic Sea from the EMODnet Chemistry database (SMHI, 2018). From the original dataset, we have extracted 1 376 674 measurements, which met the following conditions: 1) time range of 1966 - 2005; 2) spatial range of the model domain, excluding coastal observations, which fell outside the model grid; 3) S and T values exist simultaneously; 4) S is in the range of 0 ... 35 g kg$^{-1}$; 5) T is in the range of -2.5 ... 30 °C.

A preliminary check of the spatial and temporal distribution of the validation data is done. The spatial density of the data is presented on the 25 km$^2$ grid (Fig. 1a). Spatially, there are only a few horizontal cells of 25 km$^2$ that do not have any measurements. Vertically, the number of measurements decreases monotonically from the surface to the bottom following the hypsographic curve of the Baltic Sea (Jakobsson et al., 2019) (Fig. 1d). The measurements at the standard depth stick out from the overall curve. Since the end of the 1980s, the number of monthly measurements increased continuously more than an order of magnitude compared to the preceding period (Fig. 1b). Seasonally, the number of winter and early spring measurements is smaller than the number of summer measurements (Fig. 1c), which is consistent with seasonal ice coverage of the Baltic Sea (Raudsepp et al., 2020). This complicates data collection.

### 2.4 K-means clustering

The K-means clustering algorithm is a widely used algorithm in unsupervised machine learning (Jain, 2010). We use a K-means clustering algorithm for the cluster analysis of temperature and salinity errors. In the current study, two dimensional error space is defined from simultaneous salinity and temperature errors $\{dS,dT\} \in R^2$, where $dS \equiv (Smod - Sobs)$ and $dT \equiv (Tmod - Tobs)$. In general, the method can be extended to the n-dimensional error space. The distribution of the errors in the $\{dS,dT\} \in R^2$ error space is presented in Fig. 2a. Before calculating K-means, the error space has been normalized by the standard deviation of temperature and salinity errors.

The first step of the method is to determine the number of clusters. To maintain a deterministic structure of the cluster, a regular pattern of initial centroids was chosen for this study (Fig. 2b). When we start with only one cluster, we can choose its location at $\{dS=-1,dT=-1\}$. Using two clusters means that we start with the locations corresponding to 1 and 2 marked on Fig. 2b. With the increasing number of clusters, we use corresponding initial locations of the clusters marked with numbers 1, 2, 3, etc. Any other more advanced methods for the selection of initial centroids (Celebi et al., 2013) could be implemented just as well. The squared Euclidean distance was used as the measure of the distance between data points and the centroid coordinates of the cluster. The number of iterations was limited to 100, which ensured the convergence of the clustering algorithm. The disadvantage of the K-means clustering algorithm is the lack of a unique way for defining the optimal number of clusters. For the final selection of the number of clusters, we used the Elbow method (e.g., Bholowalia and Kumar, 2014; Yuan and Yang, 2009). The coordinates of the centroids in $\{dS,dT\}$ error space provide mean bias of the errors belonging to the cluster $k$. Standard deviations of dS and dT are calculated for the characterisation of the variability of the errors within a cluster.



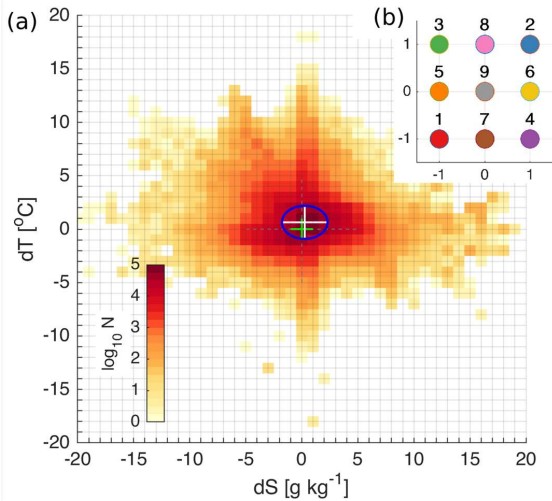

**Figure 2. Logarithmic distribution of the number of salinity and temperature error pairs (model minus observation) in the 2-dimensional error space (a). Error bins have a resolution of 1 °C for temperature and 1 g kg$^{-1}$ for salinity. The bias is shown with the center of the white cross and the standard deviations with the major semi-axes of the blue ellipse. The green cross shows the center of the coordinate axes. Coordinates of initial centroids of K-means in the normalized 2-dimensional error space (b).**

In general, the errors retain their 4-dimensional structure, i.e., {dS,dT} (t,x,y,z), while assigned to specific clusters. Any kind of analysis can be done using the clustered errors.

## 2.5 Normalization

Each error pair belongs to a fixed cluster $k$ but retains their 4-dimensional structure, i.e. {dS,dT}$^k$ (t,x,y,z). For the visualization of the model accuracy, some reduction of dimensionality of the error pairs is needed.

For the spatial distribution of errors, we take the error pairs as independent of time and vertical coordinate, i.e., {dS,dT}$^k$ (x,y). For each horizontal grid cell $(i,j)$ of 25 km$^2$, we have a number of points (error pairs) $n_{i,j}^k$ that belong to cluster $k$. The total number of points that belong to the grid cell is $N_{i,j} = \sum_{k=1}^{K} n_{i,j}^k$, where $K$ is the number of clusters. For the normalization, we divide each $n_{i,j}^k$ with $N_{i,j}$ and plot the horizontal maps for each $k$.

For vertical distribution of errors, we take error pairs as dependent only on the vertical coordinate {dS,dT}$^k$ (z). Then $n_l^k$ is the number of points in layer $l$ and cluster $k$. Total number of points in the layer $l$ is $N_l = \sum_{k=1}^{K} n_l^k$. Normalization is done for each layer with $N_l$. Subsequently, the profiles of the normalized error points show the share of each cluster of errors.

For temporal distribution of errors, we take error pairs as dependent only on time {dS,dT}$^k$ (t). Then $n_{\Delta t}^k$ is the number of points in the time interval $\Delta t$ and cluster $k$. Total number of points in the time interval $\Delta t$ is $N_{\Delta t} = \sum_{k=1}^{K} n_{\Delta t}^k$. Normalization is



done for each time interval $\Delta t$ with $N_{\Delta t}$. Then the time series of the normalized error points shows the share of each cluster of errors at a specific time.

There is no need to do normalization when we look at time series in a fixed spatial location or plot the Hovmöller diagram of
the error clusters.

## 3 Results

### 3.1 Clustering procedure

We start by clustering bulk data covering the entire modeling period and domain. Error representation does not provide a simple idea on how many clusters should be predefined or how the clusters will form. The initial location of the centroids is
selected according to the scheme shown on Fig. 2b. The coordinates of the centroid of one cluster (Fig. 3a) provide a model bias of 0.64 °C for temperature and 0.26 g kg$^{-1}$ for salinity (Table 1). Corresponding standard deviations were 1.5 °C and 2.0 g kg$^{-1}$, respectively. The root-mean square difference was 1.67 °C for temperature and 2.04 g kg$^{-1}$ for salinity. The corresponding linear correlation coefficients were 0.97 and 0.95, respectively.

Increasing the number of clusters results in splitting of the error space into clusters with centroids close to the zero point
(Fig. 3). A representative structure of distribution of the errors emerges in the case of four clusters (Fig. 3d). We can confirm the choice of four clusters by implementing cluster selection criteria. The distance between points and designated centroids reduces exponentially with the increasing number of clusters (Fig, 4). The rate of distance reduction with increasing number of clusters shows local minima at $K=4$.

The $K=4$ clustering distributes 1 376 674 error data pairs into the following four clusters, each with $N(k)$ = {263230, 196615,
134326, 782503} datapoints. Cluster $k=1$ characterizes the set of errors with the basic feature of "underestimated salinity" (Table 1). This cluster is present already in the case of three clusters (Fig. 3c). Increasing the number clusters splits this cluster into two clusters (e.g., for K=9, it splits into clusters $k=1,5$). Cluster $k=2$ envelops the errors of "overestimated salinity". This cluster changes into cluster $k=4$ ($K=5$), then splits into two clusters ($K=8$) and three clusters ($K=9$). Cluster $k=3$ of "overestimated temperature" is established already in the case of three clusters. Increasing the total number of clusters
does not result in a split of the cluster. However, the centroid shifts towards high temperature bias (Table 1). The cluster $k=4$ represents "good match" of the model and measurements. The bias is about 0.4 °C for temperature and 0.6 g kg$^{-1}$ for salinity (Table 1). The standard deviations are below one for both parameters. Increasing the number of clusters results in the splitting of this cluster along the axis of temperature error, while salinity error remains small.



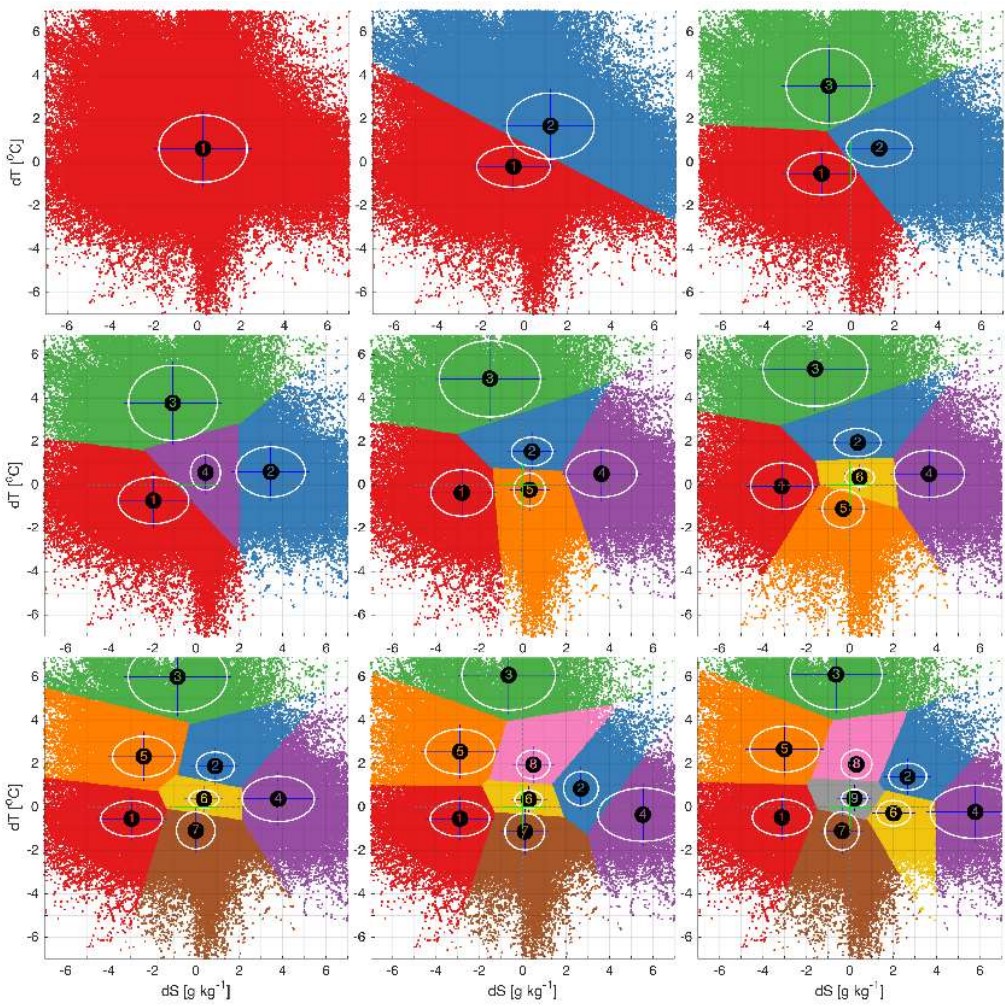

**Figure 3. The distribution of clusters in the error space for a different number of predefined clusters, *K*=1-9. The numbers of the clusters correspond to the numbers of the clusters in Table 1. The biases are marked with the center of the ellipsoid and the standard deviations with the major semi-axes. The error space has been zoomed in for better visualization of the clusters. The full range of error space and distribution of the clusters is shown in Fig. A1 in Appendix A.**





**Table 1.** The coordinates of the centroids and the standard deviations of salinity and temperature errors within the clusters for a different set of predefined clusters, *K*=1-9. The numbers of the clusters and the colors in column *k* correspond to the numbers and colors of the clusters in Fig. 3. The brighter background colors of MEAN and STD columns correspond to parental and descendant clusters of the *K*=4 cluster distribution.

| K | k | MEAN $\{dS_k, dT_k\}$ | STD $\{dS_k, dT_k\}$ | nk | k | MEAN $\{dS_k, dT_k\}$ | STD $\{dS_k, dT_k\}$ |
|---|---|---|---|---|---|---|---|
| 1 | 1 | 0.26 0.64 | 2.03 1.55 | 7 | 1 | -2.97 -0.53 | 1.42 0.82 |
|   |   |           |           |   | 2 | 0.89 1.89 | 0.88 0.66 |
| 2 | 1 | -0.49 -0.19 | 1.69 0.95 |   | 3 | -0.85 6.01 | 2.28 1.61 |
|   | 2 | 1.21 1.69 | 2.02 1.52 |   | 4 | 3.8 0.38 | 1.66 1.05 |
|   |   |           |           |   | 5 | -2.41 2.33 | 1.44 0.93 |
| 3 | 1 | -1.35 -0.51 | 1.57 0.99 |   | 6 | 0.37 0.38 | 0.69 0.42 |
|   | 2 | 1.3 0.66 | 1.52 0.85 |   | 7 | -0.01 -1.07 | 0.89 0.85 |
|   | 3 | -1.03 3.54 | 1.97 1.73 |   |   |           |           |
|   |   |           |           | 8 | 1 | -2.9 -0.53 | 1.37 0.82 |
| 4 | 1 | -1.96 -0.72 | 1.63 1.07 |   | 2 | 2.67 0.87 | 0.8 0.78 |
|   | 2 | 3.44 0.6 | 1.59 1.16 |   | 3 | -0.66 6.09 | 2.15 1.62 |
|   | 3 | -1.07 3.78 | 2.04 1.73 |   | 4 | 5.55 -0.35 | 2.09 1.23 |
|   | 4 | 0.44 0.57 | 0.69 0.81 |   | 5 | -2.89 2.56 | 1.6 1.04 |
|   |   |           |           |   | 6 | 0.27 0.36 | 0.64 0.42 |
| 5 | 1 | -2.81 -0.37 | 1.42 1.07 |   | 7 | 0.09 -1.12 | 0.91 0.85 |
|   | 2 | 0.42 1.54 | 0.95 0.66 |   | 8 | 0.48 1.97 | 0.77 0.67 |
|   | 3 | -1.52 4.89 | 2.33 1.76 |   |   |           |           |
|   | 4 | 3.63 0.52 | 1.63 1.08 | 9 | 1 | -3.13 -0.47 | 1.37 0.83 |
|   | 5 | 0.3 -0.22 | 0.72 0.77 |   | 2 | 2.67 1.41 | 0.87 0.62 |
|   |   |           |           |   | 3 | -0.63 6.12 | 2.12 1.62 |
| 6 | 1 | -3.13 -0.06 | 1.43 1.07 |   | 4 | 5.79 -0.22 | 2.08 1.23 |
|   | 2 | 0.36 1.95 | 1.08 0.65 |   | 5 | -3.01 2.68 | 1.59 1.05 |
|   | 3 | -1.59 5.35 | 2.41 1.72 |   | 6 | 2.02 -0.27 | 0.78 0.59 |
|   | 4 | 3.66 0.51 | 1.63 1.08 |   | 7 | -0.36 -1.09 | 0.79 0.88 |
|   | 5 | -0.3 -1.1 | 0.96 0.87 |   | 8 | 0.31 1.98 | 0.72 0.67 |
|   | 6 | 0.46 0.34 | 0.68 0.44 |   | 9 | 0.22 0.4 | 0.6 0.41 |





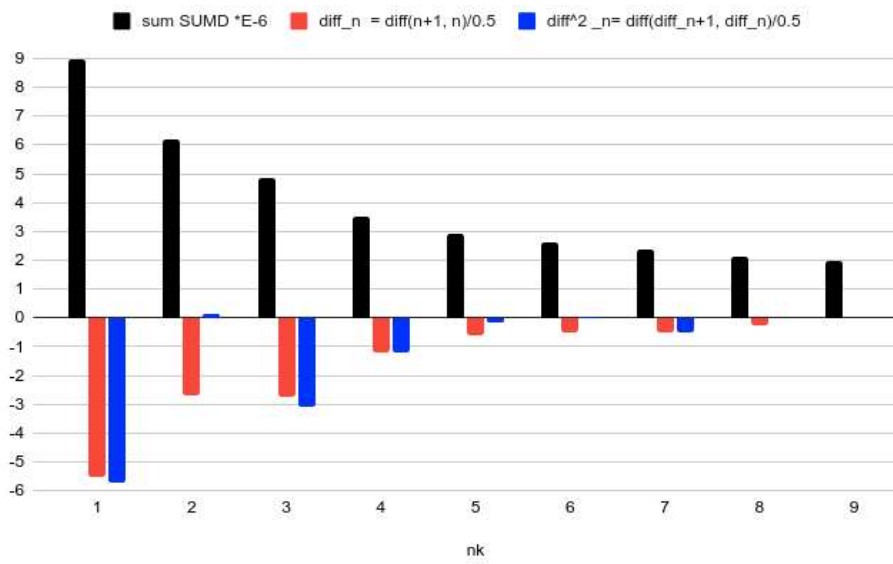


**Figure 4. Sum of square distances (black bars) between normalized pairs of error points and their designated centroids for different numbers of initial centroids. The first (red bars) and the second (blue bars) order forward differences calculated from the sum of square distances.**

### 3.2 Analysis of the clusters

Retrieving spatial coverage of $K$=4 cluster errors shows that the model has "good match" in the whole model domain (Fig. 5b). The share of the other errors remains less than 0.3. The model "overestimates salinity", "underestimates salinity" and has "good match" at the Danish straits. "Underestimated salinity" errors have a share of about 0.2 in the deep basins of the Baltic proper, i.e., the Bornholm Basin, Gdansk Basin, eastern Gotland Basin, northern Baltic Proper, western Gotland Basin and western Gulf of Finland. Model "overestimates temperature" at the transition area between the northeastern Baltic

proper and the Gulf of Finland, in some coastal locations and within the river plumes. The latter indicates that river water temperature is overestimated in the present model implementation.

Vertical distribution of the error clusters confirms that the share of "good match" errors ranges between 0.5 and 0.9 of all data (Fig. 5e). In the surface layer, in almost 50% of cases we have "overestimated salinity" and "underestimate salinity". In comparison with horizontal distribution of errors, a large part of these errors probably belong to the Danish straits (Fig. 5b).

The "overestimated temperature" has a considerable share centered at a depth of 25 meters. The "underestimated salinity" has a high share at the depth range of 60-100 m. The share of "underestimated salinity" once again increases in the deep layer of the Baltic Sea.



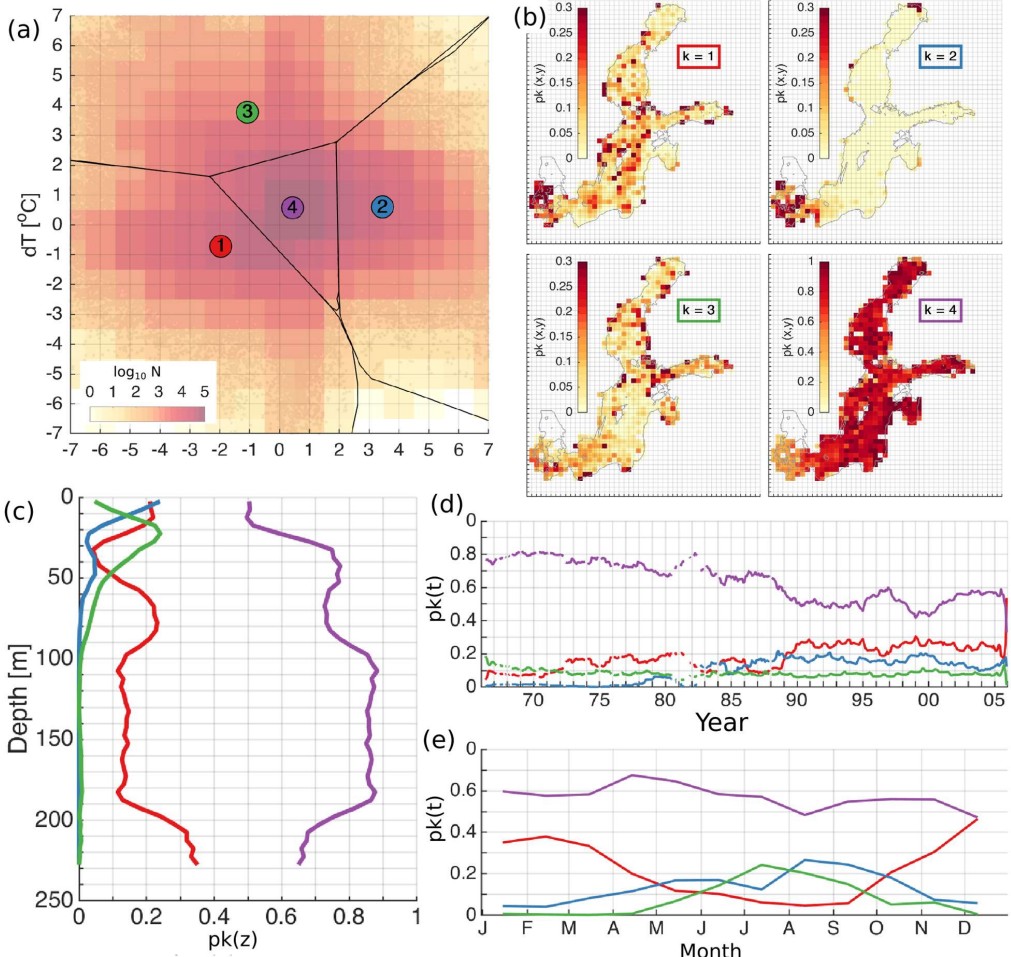

**Figure 5.** The distribution of the error clusters for *K*=4 (a). The colormap shows the logarithmic distribution of the number of salinity and temperature error pairs (model *minus* observation) in the 2-dimensional error space (a). Error bins have a resolution of 1 °C for temperature and 1 g kg⁻¹ for salinity (a). The spatial (b), vertical (c), temporal (d) and seasonal (e) distribution of the share of error points belonging to the four different clusters (b). The share is calculated as explained in Section 2.4. The horizontal bins have a resolution of 25x25 km (b), vertical bins have a resolution of 5 m (c), temporal and seasonal bins have monthly resolution (d,e). The lines (d) have been smoothed using a running mean with a 12-point window size. Line colors correspond to the colors of the clusters on (a).

A decrease in time of a "good match" coincides with an increase of the share of "underestimated salinity" and "overestimated salinity" (Fig. 5c). Seasonally "overestimated salinity" has a higher share in summer, while "underestimated salinity" has a higher share in winter (Fig. 5d). Combining horizontal (Fig. 5b) and seasonal distribution of errors (Fig. 5d),



we could conclude that the salinity is overestimated in the Danish straits in summer and underestimated in winter. In
addition, we would like to note that the share of "good match" decreases and "underestimated salinity" increases abruptly at
the end of the 1980s, when the number of the measurements becomes larger in the database. The "overestimate temperature"
has an almost constant share of 0.1 in time (Fig. 5c). The elevated share of "overestimated temperature" errors in summer
confirms that the model overestimates the temperature in the seasonal thermocline (Fig. 5d). For comparison, we have
provided a similar analysis of the errors for $K$=3 and $K$=5 in the Appendix B.

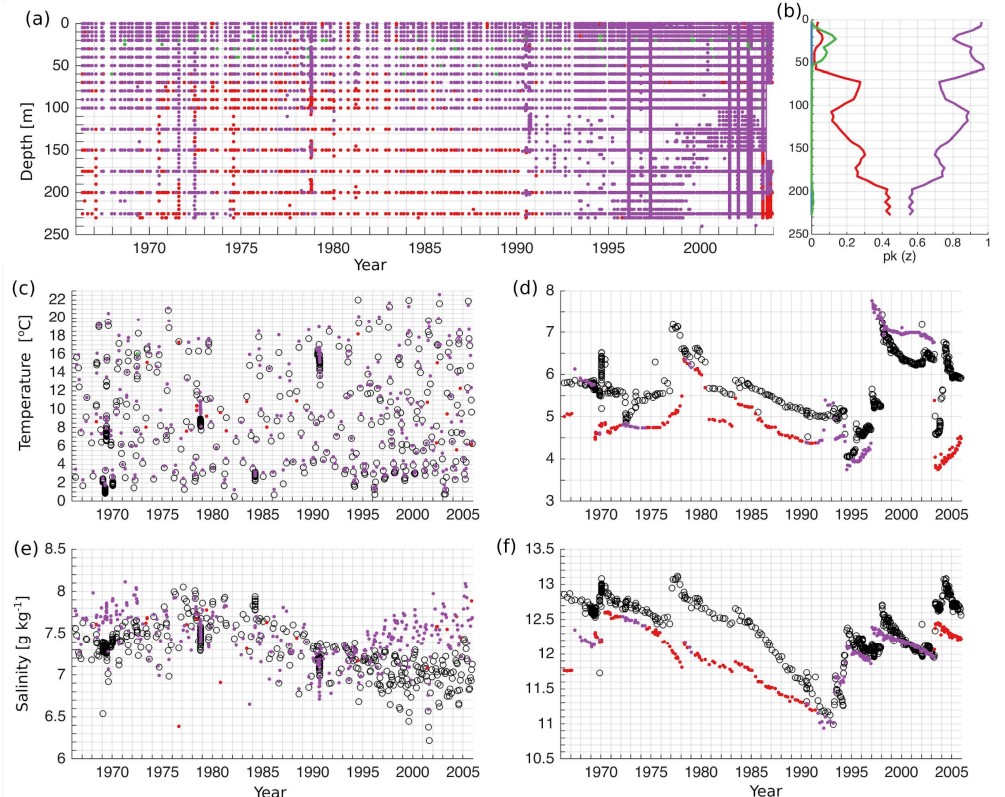


**Figure 6  Hovmöller diagram of the distribution of error points of $K$=4 at the BY15 monitoring station (a). Vertical distribution of
the share of error points belonging to the four different clusters (b). Time series of observed (black circles) and simulated (color
dots) temperature (c,d) and salinity (e,f) on the surface (c,e) and bottom (d,f) at the BY15. Colors of dots and lines correspond to
the colors of the clusters on Fig. 5a.**

We extract error profiles from Gotland Deep station BY15, which is widely used for the validation of the physical and
biogeochemical models of the Baltic Sea. In the upper layer of 60 m, the model has "good match" (Fig. 6a,b). There are
isolated occasions of 10% in total, when the model "overestimates temperature" in the seasonal thermocline (Fig. 6b). At the



depth range 60-100 m, the share of model "underestimating salinity" increases. From a depth of 100 m, the proportion of the model that "underestimates salinity" gradually increases with depth. The Howmüller diagram shows that there are extended

time periods when the model "underestimates salinity" (Fig. 6a). In the surface layer, the model has "good match", although model salinity starts to deviate from the measurements starting from 1995 (Fig. 6c,e) . At the bottom, the model reproduces temperature very well at the end of 1970s and beginning of 1980s, but as salinity is underestimated, the errors belong to the cluster of "underestimated salinity" (Fig. 6d,f). In general, the model has "good match" in the water column from 1991 to 2003 (Fig 6a,f). Dynamically, this corresponds to the end of the stagnation period and recovery of the bottom salinity and

strengthening of the permanent halocline.

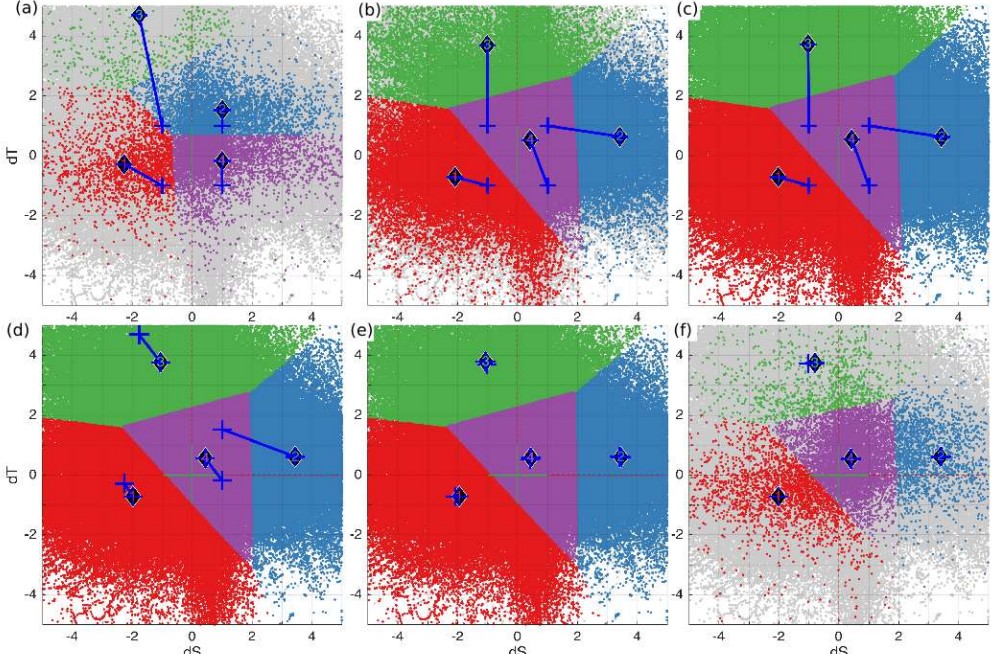

**Figure 7. Learning (a-c) and predicting (d-f) of the *K*=4 clusters. The learning and predicting datasets have a share of 1% (a) and 99% (c), 20% (b) and 80% (e), 99% (c) and 1% (f) of the full dataset, respectively. Blue crosses mark the location of initial centroids and blue lines connect initial and final locations (marked with numbered diamonds) of the centroids.**

**3.3 Learning of the clusters**

As the first step, the whole 4-dimensional {dS,dT} dataset is divided randomly into two separate sets for learning and predicting. The dataset for the learning of the error clusters is initiated from a set of a different number of clusters according to initial distribution of the centroids shown on Fig. 2b. Resulting centroids of the learning dataset are then used to initiate



the centroids for the clustering of the predicting dataset. The mean length of shifts between learning and predicting centroids
is used to evaluate the effect of dataset size on predicting the representative error clusters. We have used different learning
and predicting datasets with sizes ranging from a share of $10^{-4}$ to 0.9999 of the total dataset of 1 376 674 error pairs. The
average distances are calculated from 30 trials to have a statistical ensemble of randomly selected datasets. The learning and
predicting procedure is illustrated in Fig. 7 for $K$=4.

If the learning dataset makes up 10-95 % of the total dataset (>100 000 comparison points), then the difference between the
learned and predicted centroids does not change significantly (Fig. 8). The clustering of $K$=4 is most sensitive to the choice
of initial centroids. Therefore, the distance between learned and predicted centroids is larger compared to other choices of K.
Below 1% of the learning data size (<10 000 comparison points), the difference in distance between learned and predicted
datasets is >0.03 normalized standard deviation. Thus, the size of the learning dataset is significant for predicting the error
clusters. The rough estimate of the number of comparison points is about 100 000 for the current model, which ensures
relatively stable centroids.

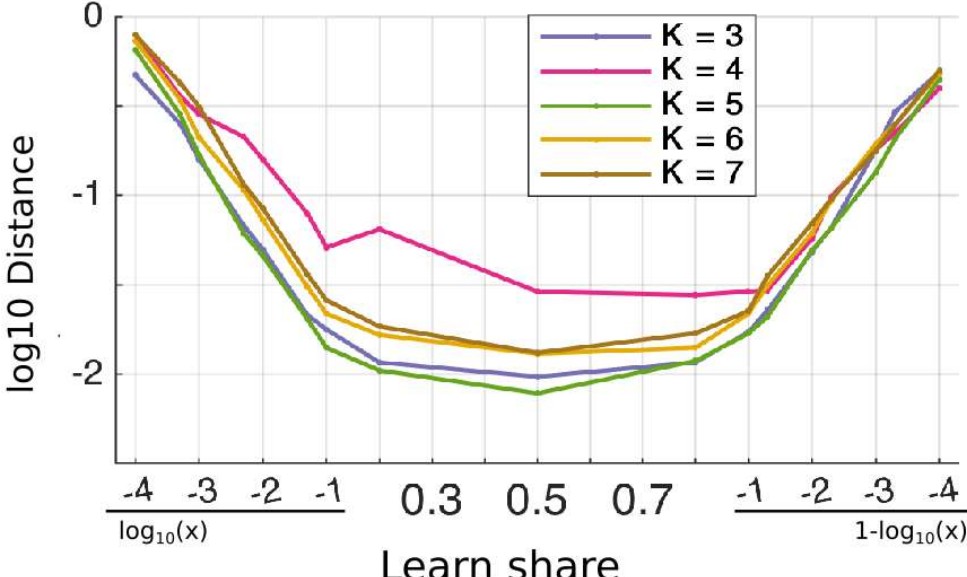

**Figure 8. The average normalized distance of shifts of predicting centroids relative to learned centroids as function of the share of the learning dataset. Averaging has been done from 30 trials. Different lines correspond to different numbers of initial clusters, $K$=3-7. The share of the learning dataset in the ranges of $10^{-4}$-0.1 and 0.9-0.9999 are shown in logarithmic scale.**





### 3.4 Interpretation of the clusters


Total number and the spatio-temporal coverage of the comparison points (Fig. 1) indicates that the model performs well over the Baltic Sea and the simulation period considered (Fig. 5). The share of model errors with a bias of {dS,dT}={0.44 g kg$^{-1}$,0.57 °C} and with a standard deviation of {dS,dT}={0.69 g kg$^{-1}$,0.81 °C} (Table 1) is between 0.5 and 0.9.

In addition, we can highlight the areas where the model accuracy is lower and the dynamical features are not so well

reproduced by the model. Essentially, seasonal thermocline and permanent halocline are not reproduced by the model as well as the layers with small vertical gradients of salinity and temperature. The model accuracy in reproducing seasonal thermocline has a peak share of "overestimated temperature" of 0.25 (bias of 3.78 °C and standard deviation of 1.73 °C) at a 25 m depth. The error share of 0.25 is observed in the layer of 60-90 meters, which corresponds to the depth range of the permanent halocline. The model "underestimates salinity" (bias of -1.96 g kg$^{-1}$ and standard deviation of 1.63 g kg$^{-1}$) there.

The model accuracy is relatively low in the Danish straits. The model has "underestimated salinity" in winter and "overestimated salinity" in summer (bias of 3.44 g kg$^{-1}$ and standard deviation of 1.59 g kg$^{-1}$) there. The "underestimated salinity" errors in the deep basins of the Baltic Sea (Fig. 5b) are caused by the spreading of inflowing North Sea water downstream in the cascade of the deep basins. These inflows mainly take place in winter, while outflow of the Baltic Sea water dominates in summer.

Clustering of model errors could provide information about the accuracy of external fields that are used for the forcing and for the boundary conditions of the model. The "overestimated temperature" at the river plume areas (Fig. 5b) may indicate a mismatch of river water temperature that takes the value from a grid cell adjacent to the river mouth. Although the air-sea fluxes are correctly reproduced by the model, as indicated by "good match" at the surface (Fig. 5c), the following downward flux of heat could be too strong, as the share of "overestimated temperature" is relatively high between the depth of 10-40

meters in summer (Fig 5c,d).

### 4. Summary

Ideally, researchers like to know the model accuracy over the whole model domain and time period simulated. Commonly used methods provide a limited set of metrics (e.g., bias, standard deviation, root mean square error, correlation coefficient) for the assessment of the model overall quality. In this study, we have proposed a new method for the assessment of the

model skills. The aim of using the method is the clustering of multivariate model errors. Model errors consist of differences between the model values and the measured multivariate data. The main advantage of this method is the possibility to use clustered errors for the analysis of the spatio-temporal accuracy of the model.

The method was tested in the validation of the circulation model results of the 40-year period in the Baltic Sea. Temperature and salinity were used for the validation, because they are essential parameters of the physical model and these data have

been the most extensively measured in the Baltic Sea. This method enables us to use all available observations, with the only restriction being that multivariate data has to be measured simultaneously. In model validation, the problem usually lies in



the spatio-temporal distribution of measurement data over the 4-dimensional model domain. In our case, the measurement data was sufficient and with good spatial and temporal coverage. In total, we had more than 1 300 000 pairs of measured temperature and salinity values. In many cases, reduction of available data or homogenization of the data is needed prior to

the calculation of model errors, and clustering is applied to have simultaneous multivariate data. The number of measurements should be sufficiently large to determine stable clusters. In our case, about 100 000 randomly selected data pairs ensured the stability of the clusters.

We have applied the K-means unsupervised machine learning algorithm for the assessment of the quality of general circulation models by clustering the temperature and salinity errors. The model output fields are 4-dimensional, and the 4-

dimensional distribution of the errors was retained after the clustering was completed. As a result, cluster numbers were assigned to each error pair. In addition, the errors belonging to one cluster had their bias determined by the location of the centroid in the error space. Further on, common statistical metrics (e.g., standard deviation, root mean square error, correlation coefficient) can be calculated for each cluster and variable. In general, any other partitional clustering algorithm can be used instead of K-means for the clustering of multivariate model errors. We have implemented the K-means

algorithm because of its simplicity and robustness. The outcome clusters have direct information of the model bias. The output clusters can be used for calculation of classical statistical metrics. Resulting clusters contain information about common statistical metrics.

The K-means clustering algorithm has a well-known deficiency. There is no unique way to determine the number of clusters. We used Elbow methods, which gave good results. The selection of four clusters was supported by the analysis of the error

clusters in relation to the geographical distribution of the errors, the physical process and the features. The analysis showed that the "underestimated salinity" cluster was mainly in the Danish straits, within the halocline layer and along the pathway of transport of saline water in the Baltic Sea. "Overestimated temperature" had a high share in the seasonal thermocline. "Overestimated salinity" accounted for the model errors in the Danish straits. For confidence, the analysis was complemented with using three and five clusters. Thus, the analysis of the error clusters enables to shed light on the physical

processes and features where model performance should be improved.

The clustering was done for the entire Baltic Sea and the whole simulation period. Analysis of clusters of errors at specific locations enables us to assess the model quality there in the context of the overall quality of the model. Multivariate model quality assessment shows that if one parameter is well reproduced by the model, but the other parameter is poorly reproduced at the same time, then the quality might not be good and vice versa.

In addition to model quality, error clustering can provide implicit information about the quality of prescribed input variables and forcing fields. Error clustering has shown that the temperature of river runoff water could be overestimated. This is especially relevant for the biogeochemical models, where discharges of different nutrients and other state variables, which have to be prescribed, are usually poorly known. There are problems in the prescribed salinity of the inflowing North Sea water at the model open boundary in the Kattegat. In addition, these errors are transported into the model domain of





southwestern Baltic Sea. On the other hand, atmospheric fields necessary for the calculation of the air-sea heat fluxes do not produce significant errors.

The proposed method could be applied for the assessment of the quality of global ocean general circulation models. By the end of the year 2020, there were approximately 3800 ARGO floats profiling the world ocean for salinity and temperature, with a spatial resolution of approximately 1 float for every 3 degrees of latitude and longitude. The annual total number of

profiles added to the database is over 100 000, which takes the total available number of profiles to over 2 000 000 (Argo, 2020). This huge validation data set probably needs some computational solution, i.e., implementation of parallel computing or specific methods on how to deal with big data within the K-means clustering. In the context of operational oceanographic models, the model validation can be done in "real time" by implementing the learning-predicting sequence. The ARGO data, which are available within 24 hours of collection, could be added to the learned clusters for the updating of the coordinates

of the centroids and statistical metrics.

The proposed method can be applied to different geoscientific models. The shortlist consists of biogeochemical models, atmospheric models, wave models, hydrological models, geodynamic models. The method can be implemented in a multivariate high-dimensional error space as well as in a univariate error space. In addition to the validation of numerical models, the method can be used for the assessment of remote sensing data and models.



**Appendix A**

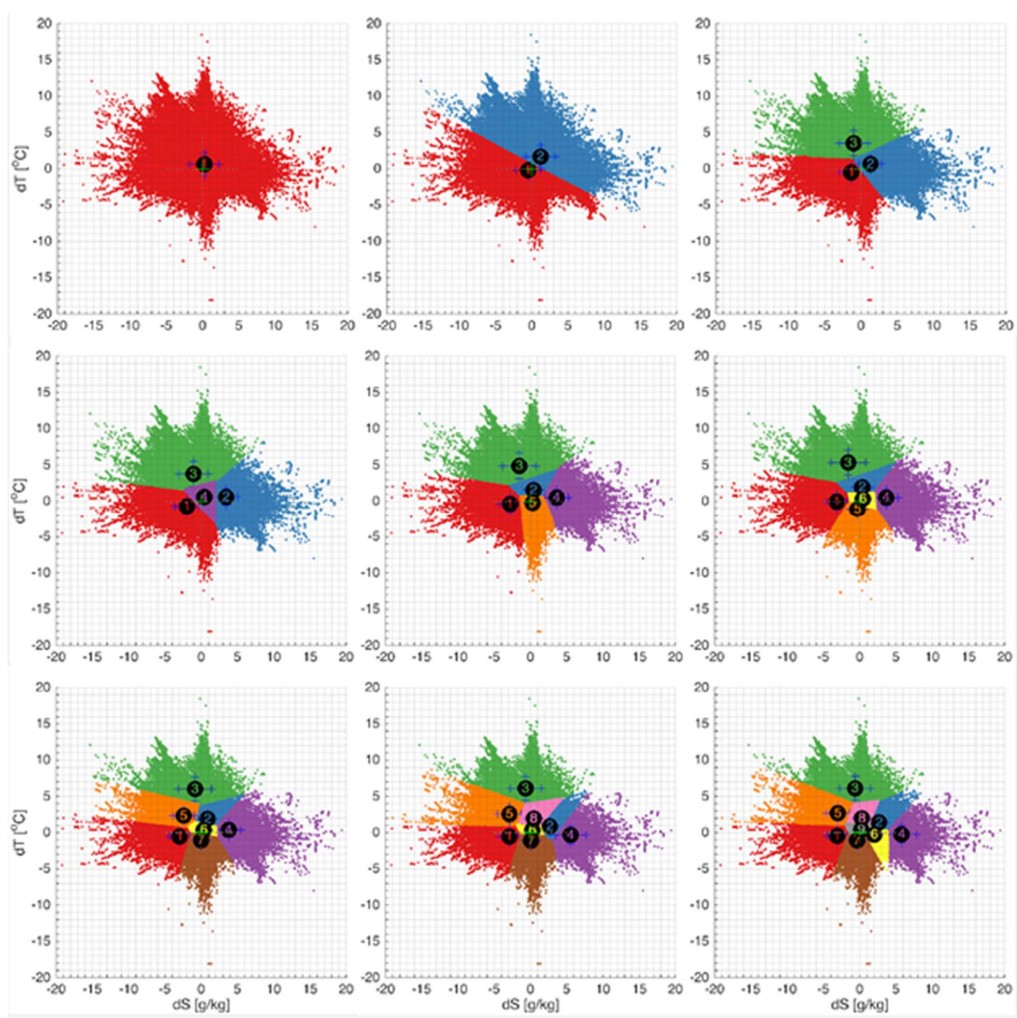

**Figure A1. The distribution of clusters in the error space for a different number of predefined clusters, *K*=1-9. The numbers of the clusters correspond to the numbers of the clusters in Table 1. The locations of the centroids are marked with cluster numbers and the standard deviations with the whiskers.**





**Appendix B**


In the case of three clusters, the largest share of errors belongs to the cluster $k=2$ with a bias of {dS,dT}={1.3 g kg$^{-1}$,0.66 °C} and with a standard deviation of {dS,dT}={1.52 g kg$^{-1}$,0.85 °C} (Fig. B1). This cluster provides the main contribution to the clusters of "good match" and "overestimated salinity" when a larger number of clusters is used. The share of the errors of this cluster is between 0.6 and 0.9. Cluster $k=1$ with a bias of {dS,dT}={-1.35 g kg$^{-1}$,-0.51 °C} and with a standard deviation

of {dS,dT}={1.57 g kg$^{-1}$,0.99°C} is the cluster of "underestimated salinity", which retains these features throughout the increasing of the number of clusters. Spatially, "underestimated salinity" has a significant share in the Danish straits and on the pathway of inflowing saline water through the deep basins of the Baltic Sea. Vertically, these errors have a large share of 0.5 in the layer of 60-110 m, which corresponds to the permanent halocline of the Baltic Sea, and below 200 m, which is the bottom layer of the Gotland Deep. The share of "underestimated salinity" is relatively high in the whole water column below

the halocline. Seasonally, these errors are significant in winter, when saline water inflows through the Danish straits to the Baltic Sea occur. Cluster 1 with a bias of {dS,dT}={-1.03 g kg$^{-1}$,3.54 °C} and with a standard deviation of {dS,dT}={1.97 g kg$^{-1}$,0.73 °C} has a steady share of errors of 0.1. The errors of "overestimated temperature" are significant in the depth range of 10-50 m and during summer. These errors account for the model accuracy in reproducing seasonal thermocline.

In the case of 5 clusters, the clusters $k=2$ with a bias of {dS,dT}={0.42 g kg$^{-1}$,1.54 °C} and with a standard deviation of

{dS,dT}={0.95 g kg$^{-1}$,0.66 °C} and $k=5$ with a bias of {dS,dT}={0.3 g kg$^{-1}$,-0.22 °C} and with a standard deviation of {dS,dT}={0.72 g kg$^{-1}$,0.77 °C} dominate over the others (Fig. B2). These clusters are formed as a split of the "good match" cluster with partial contribution from the "underestimated salinity" cluster and the "overestimated salinity" cluster of $K=4$. The clusters $k=1$ with a bias of {dS,dT}={-2.81 g kg$^{-1}$,-0.37 °C} and with a standard deviation of {dS,dT}={1.42 g kg$^{-1}$,1.07 °C} and $k=4$ with a bias of {dS,dT}={3.63 g kg$^{-1}$, 0.52 °C} and with a standard deviation of {dS,dT}={1.63 g kg$^{-1}$,1.08 °C}

share errors of "underestimated salinity" and "overestimated salinity". These errors dominate in the Danish straits, indicating the model difficulties in matching fluctuating water salinity close to the model boundary. Cluster $k=3$ with a bias of {dS,dT}={-1.52 g kg$^{-1}$,4.89 °C} and with a standard deviation of {dS,dT}={2.33 g kg$^{-1}$,1.76 °C} accounts for "overestimated temperature" errors in the seasonal thermocline during summer.


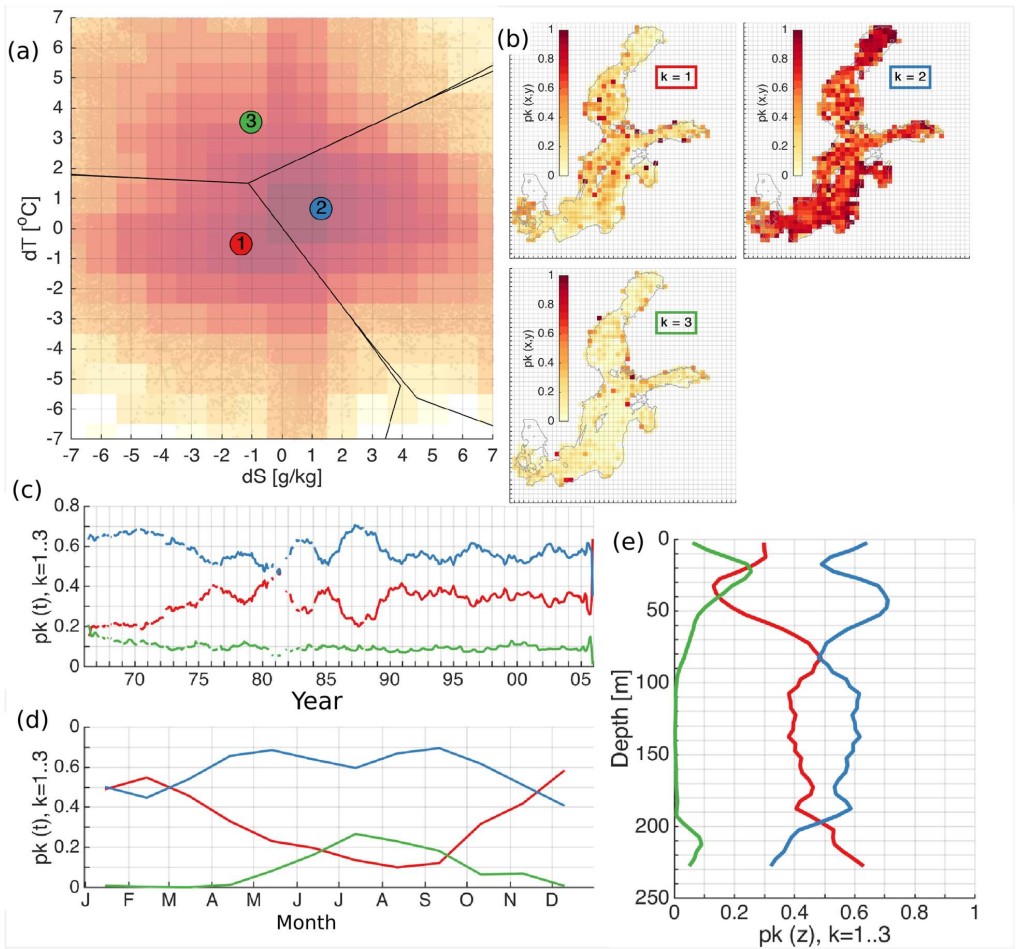

**Figure B1.** The distribution of the error clusters for *K*=3 (a). The colormap shows logarithmic distribution of the number of salinity and temperature error pairs (model *minus* observation) in the 2-dimensional error space (a). Error bins have a resolution of 1 °C for temperature and 1 g kg⁻¹ for salinity (a). The spatial (b), vertical (c), temporal (d) and seasonal (e) distribution of the share of error points belonging to the four different clusters (b). The share is calculated as explained in Section 2.4. The horizontal bins have a resolution of 25x25 km (b), vertical bins have a resolution of 5 m (c), temporal and seasonal bins have monthly resolution (d,e). The lines (d) have been smoothed using a running mean with a 12-point window size. Line colors correspond to the colors of the clusters on (a).





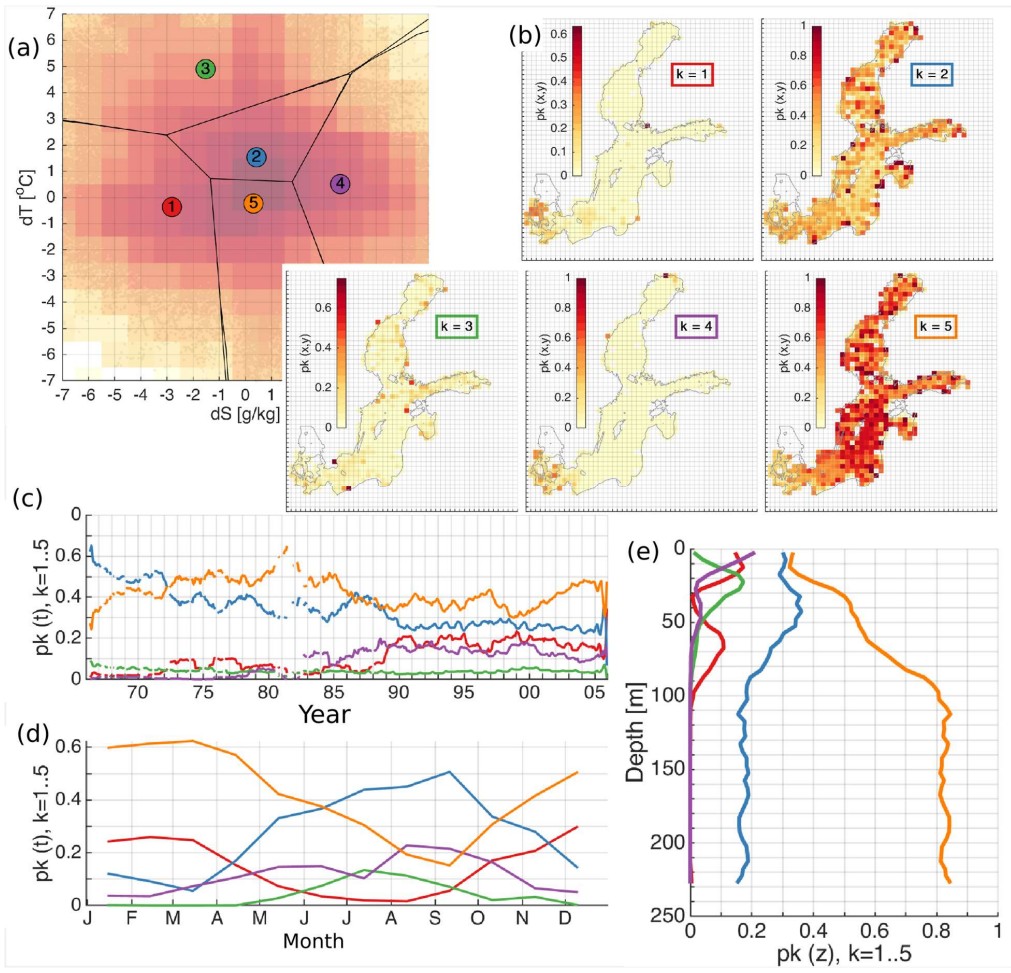

**Figure B2.** The distribution of the error clusters for *K*=5 (a). The colormap shows logarithmic distribution of the number of salinity and temperature error pairs (model *minus* observation) in the 2-dimensional error space (a). Error bins have a resolution of 1 °C for temperature and 1 g kg⁻¹ for salinity (a). The spatial (b), vertical (c), temporal (d) and seasonal (e) distribution of the share of error points belonging to the four different clusters (b). The share is calculated as explained in Section 2.4. The horizontal bins have a resolution of 25x25 km (b), vertical bins have a resolution of 5 m (c), temporal and seasonal bins have monthly resolution (d,e). The lines (d) have been smoothed using a running mean with a 12-point window size. Line colors correspond to the colors of the clusters on (a).

**Data availability**

Data used in this article are available online at https://zenodo.org/record/4588510#.YIjni6hRW-M



 (Maljutenko, 2021). The data is error space supplemented with time and space coordinates and cluster indexes for K=1-9. Fur clustering the K-means function from Statistics and Machine Learning Toolbox of MATLAB R2020a was used.

**Competing interests**

The authors declare that they have no conflict of interest.

**Author contributions**

Following tasks were done by the authors

UR: Conceptualization, Methodology, Writing – original draft preparation, Formal analysis.

IM: Software, Visualization, Data curation, Formal analysis.

**Acknowledgements**

This study was financially supported by the European Regional Development Fund within the National Programme for Addressing Socio-Economic Challenges through R&D (RITA1/02-52-04). We would like to thank Dr. Rivo Uiboupin and Prof. Jüri Elken for valuable comments. A special thanks to Miss Meelimari Aljasmäe from Urmas Raudsepp for support during the preparation of the manuscript.

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
