# Peer review of "A method for assessment of the general circulation model quality using K-means clustering algorithm: a case study with GETM v2.5"

_Geoscientific Model Development, 2021_

## Referee Comment (RC1)

**REFEREE REPORT:**
**A METHOD FOR ASSESSMENT OF THE GENERAL CIRCULATION MODEL QUALITY USING $K$-MEANS CLUSTERING ALGORITHM**
**U. RAUDSEPP, I. MALJUTENKO**

This paper suggests using $K$-means as a method for clustering error estimates of the General Estuarine Transport Model (GETM), an oceanic general circulation model, to find meaningful spatial structure of model error. Error estimation is done by taking the difference between true values of daily averaged temperature and salinity found in the EMODnet Chemistry database and simultaneous values generated by the GETM for a 3 dimensional spatial region $(x, y, z)$ around the Baltic sea and time $t$. Once error values for temperature $dT = dT(x, y, z, t)$ and salinity $dS(x, y, z, t)$ are found, their unique pairs $(dT, dS)$ are plotted in the $\mathbb{R}^2$ clustering space. $K$-means is then performed using the euclidean metric in $\mathbb{R}^2$. Since each pair is uniquely identified by $(x, y, z, t)$, the clustering result can be evaluated in the original (Lat, Lon) space and time.

It is obvious that the authors have a well-established understanding of the dynamics of ocean flow and the available models in this field as shown in section 2.1-2.2. Furthermore, the idea of using error structure to determine where possible biases exist geographically is meaningful. However, there are several key aspects of the method that has been introduced that lead to invalid or uninterpretable results of $K$-means. Because of this, I do not believe at this time that the study is complete enough for publication. I list the reasons below. Additionally, I would suggest the authors look into some statistical learning literature such as The Elements of Statistical Learning, by Jerome H. Friedman, Robert Tibshirani, and Trevor Hastie.

1. **The position of initial centroids for $K$-means is uniformly distributed and does not change in the paper.** The $K$-means algorithm is heavily dependent on the choice of initial centroids. Hence, the algorithm may reach a local minima for the distance metric but it cannot be said that this minima is the global minima (or the "most optimal"). To overcome this dependence, the $K$-means algorithm is often run many times with randomly placed centroids. If the clustering outcome remains consistent, then it is seen as a relatively reliable outcome for the algorithm.

2. **There is no obvious "elbow" in the plot of the distance.** Although this step can be thought of as suggestive, an elbow in the data should correspond to the number of clusters $K = k$ such that the change in rate forms a jump in the rate function at $K = k + 1$. If the plot of minimum distance over $K$ is connected across the histograms in figure 4, it has a smooth exponential decay which indicates that there is no obvious cluster structure for $K$-means.

3. **Halting the $K$-means process after 100 iterations does not "ensure convergence".** Line 141 states that the number of iterations is limited for the numerical implementation of $K$-means to 100. This is not good numerical practice as it is not guaranteed that you will reach the local minima for the set of initial centroids in this number of iterations. Instead, it is numerically common to set a threshold for example, $O(10^{-4})$ so that if the change in the distance metric is less than this for a given number of iterations then it is assumed to converge.

4. **$K$-means requires the clusters in the space to be able to be divided by hyperplanes (lines in the case of $\mathbb{R}^2$) and the distribution of error makes this difficult.** Contrary to the statement made on line 173, error representation can actually provide some insight on the possible structure of clusters in the $\mathbb{R}^2$ error space. It is common in statistics to assume that the error of a model (like that of temperature and salinity in this case) is the sum of many independent errors, which by the central limit theorem tend to the normal distribution with mean zero and variance $\sigma^2$. Of course there are exceptions to this in the case of dependence or outliers. Because we can reasonably assume a normal distribution for the errors of both temperature and salinity, we know if we plot these values on $\mathbb{R}^2$ that the majority of the points will be in the center with the number of observed points away from the origin decaying by the variance of their normal distribution. For an illustration, see a projection of the bivariate Gaussian distribution. This is also seen in figure 2.

   The reason I mention the normality of the error distributions for temperature and salinity is because it is not possible to appropriately divide this type of resulting cluster structure by hyperplanes unless you increase the number of clusters $K$ to some very large amount. In fact, the approximation of a circular cluster with a center can be seen in figure 3 as the number of clusters $K$ increases. Given this, I would suggest that the authors look at other possibilities of clustering such as kernel $K$-means where the divisions can be made in a functional space. This may also give some reason as to why there is no obvious elbow in the distance vs. $K$ graph.

5. **$K$-means clustering does not add any information on the structure of the data. Many of the results in section 3.2 can be found without clustering.** The spatial locations provided in figure 5 and discussed in section 3.2 where over- and under- estimates of temperature and salinity occur in the model can be found by setting a threshold of over- and under- estimation, say 2 standard deviations away from the mean, for each measurement and calculating the proportion of points in the (lat, lon) space that fall above or below the set thresholds.

6. **The process described in 3.3 just describes the continuation of the $K$-means algorithm.** The $K$-means algorithm begins with a set of random centroids, assigns points to the centroids based on their proximity, recalculates the centroids based on the mean, and continues this way until a local minima of the inter-cluster distances is reached. If the algorithm is run on a uniformly selected subset of points coming from a fixed distribution (which is true in this case), the reassignment will

continue until a local minima is obtained. If more points coming from the same distribution are added and the end centroids are used, the algorithm will continue from its final centroids until it reaches the exact clustering structure that is unique to the starting centroids, this is the case with figure 7 (a) (d). If less points are added, the structure remains constant (but the same since it corresponds to the same starting centroids), this is the case with figure 7 (b) (e) and (c) (f). This does not provide information on the *stability* of the clusters (line 274-275) which depends on changing the starting centroids and performing multiple runs of the $K$-means algorithm.

Additionally, there are some other minor issues with the article such as grammatical issues in the switching of "the" and "a" as in the first sentence of the abstract, present tense writing should be used when describing the work done for this article, and some spelling issues (e.g. line 50). There is a lack of literature on clustering methods with Jain (2010) being the main reference. As I mentioned, I would suggest starting with the book by Friedman, Tibshirani, and Hastie for a foundational understanding of unsupervised learning algorithms.

---

## Referee Comment (RC2)

Review of manuscript

"A method for assessment of the general circulation model quality using K-means clustering algorithm"

by Urmas Raudsepp, Ilja Maljutenko

General comments:

Including machine learning approaches for multi-model validation has been a long-standing discussion in CMEMS. For over ten years, the four clusters validation approach has been used as a general guideline. I'm delighted to see that progress is being made with machine learning methods. Thus, there is no doubt that the subject is worthwhile of investigation. However, the study given in this manuscript did not convince me to utilize this method directly, for example, in the CMEM QUID report. My primary worries are as follows:

1. Clustering techniques are frequently used to evaluate atmospheric models, biogeochemical models, and so on. The variables in those models are multidimensional and, to an extent, "colossal." Typically, the output of an ocean circulation model is not regarded as a massive dataset. To persuade me to experiment with various clustering approaches based on machine learning, the interpretation of the clusters should be striking.

2. Prior impacts on clustering approaches, particularly hierarchical clustering methods, should be acknowledged. Without a doubt, comparing hierarchical clustering against centroid-based clustering is worthwhile.

3. The cluster interpretation should emphasize the distinct outcomes using the Taylor and target diagrams. At the moment, I see no evidence of new information being obtained (my last comment).

4. The Baltic Sea is very special. The salinity is significantly lower than that of other marginal seas, and interaction with the open ocean is extremely limited, among other factors. I have my doubts about the method applied to the Baltic Sea being universally applicable; yet, this should be discussed.

As a result, I recommend that the authors pursue two revision strategies for the paper. One possibility is to include more model data (sea level, mixed layer depth, currents, sea ice, and possibly heat fluxes and runoffs), or to use multiple models (at least two, another one can be CMEMS results). This way, I can determine the method's reliability. Another possibility is to incorporate additional clustering methods, such as agglomerative hierarchical clustering (bottom-up), divisive hierarchical clustering (top-down), or 'soft' K-means clustering (distribution-based) vs. rule-based methods (geographic areas, etc.). Clustering evaluation enables the acquisition of beneficial best-practices for clustering analysis. I believe that the work in these two areas does not require much time, and hence I recommend a major revision.

P-Page, L-Line

Introduction:

P3, L40-L41: The rationale for using clustering methods is unclear. The shortcoming is that those papers did not include enough information in data? What is 4 dimensional information embedded? For instance, vertically, even if the vertically resolution in the observation is 1 cm, but you still bin to the resolution of 5m, don't you? You did not include more information than traditional methods. I feel that the problem of standard statistical metrics (Taylor and target diagram) is their inability to express clustered error statistics, such as error in climatology, seasonal, or diurnal signals. By the way, what are your criteria for defining 'the huge dataset'?

P3, L49: It appears as though this 'K-means clustering algorithm' has fallen from the sky. This section should contain an introduction to conventional clustering algorithms. There is something missing at the start of L50.

P3, L60-64: This section should be in the 'discussion or perspective'. Why in the 'introduction'? Perhaps some previous efforts have already made used of it in an operational mode? Then they should be cited.

P3, L68-70: This article discusses the results for the entire Baltic Sea. Other validation studies of GETM in the Baltic Sea, not just in the Gulf of Finland, should be cited.

Materials and Methods:

P4 Why this subsection 2.1 is in 'Methods'? It should be in the introduction part, and review of the Baltic Sea dynamics should be included, with a reference to the discussion in the subsequent section on 'adopting this method with caution' in other seas.

P6, L120, What is meant by a 'preliminary' check? That is, by examining Fig. 1a?

P6, L127, 'This complicates data collection.' What does it mean? Perhaps you mean 'gathering of data during winter is very complicated'?

P6, L140, 'The squared Euclidean distance' is also coming from sky. Is that different clustering measures should be introduced and the reason to not choose non-Euclidean measures should be clearly stated.

Results:

P12, Figure5d, the dramatic change of clusters in recent years, e.g. big increase of K1, is it because of the smoothing you applied? BTW, add the meaning of pK in caption.

P16, Section 3.4: Interpretation of the clusters. My concern 3 reflects the issue raised in this section. Almost all of the problems in this section can be well-defined using traditional methods and have generally recommended solutions, e.g. poorly

simulated thermocline (increasing vertical resolution), Baltic inflow problem (increasing bottom inflow), Danish strait problem (too close to open boundary), river temperature problem (no easy solution), SST problem (bulk formula). Nothing novel! I would anticipate more new information if authors include more data than T and S. While one may argue that this is not critical, if not the primary need of GMD, it gives me, as a modeler, the feeling that this method is unnecessary.

---

## Author Comment (AC1)

Dear reviewer

We very much appreciate your comments on the unsupervised machine learning aspects of the model validation method. We are familiar with the book "The Elements of Statistical Learning. Data Mining, Inference, and Prediction" by Trevor Hastie, Robert Tibshirani, Jerome Friedman.

The model errors do not have obvious clusters unless there are obvious errors in the model. Our clustering strategy was to get meaningful spatio-temporal distribution of the model errors based on the distribution of data in the clusters. In addition, common statistics that have been used so far in the validation of the models can be calculated for each cluster. We wanted to keep the clustering procedure relatively simple and easy to implement for the wide audience who are not experts in the field of unsupervised machine learning. The other aspect of using the K-means, instead of the other algorithms, e.g. kernel K-means, was the relatively low computational time. Usually, in ocean model validation we deal with a huge dataset.

We can drop the idea of using uniform distribution of initial clusters, i.e. to use random clusters and run the clustering until the clustering outcome remains consistent, and use $O(10^{-4})$ criterion for the convergence. We can change the corresponding text in the manuscript. Our tests showed, while doing research and preparing the manuscript, that changes in the results are minor.

Considering your concerns:

1) We have made experiments with a) random selection of the initial centroids, b) random selection of the initial centroids in the range of min/max data rectangle and c) uniformly distributed initial centroids as described in the paper. In all cases centroids converged almost to the same locations. To use uniformly distributed initial centroids was merely suggestion to start with and check if meaningful clusters in terms of numerical model under consideration occur.

2) We agree that there is no obvious elbow in the plot of the distance. But the distance does not have a smooth exponential decay. We calculated the first and second derivative of the distance as function of the number of clusters, which suggests using of 2 or 4 clusters.

Indeed, we agree that there is no obvious cluster structure in our dataset. For us was challenging to implement clustering algorithm to the dataset where there were no obvious clusters. Our aim was to see is there meaningful clusters in the context of the application.

3) We agree that halting the K-means process after 100 iterations does not "ensure convergence". All our tests showed that K-means process converged after 100 iterations. Thus we used 100 iteration as an indicative number of iterations.

4) We agree that in case of "ideal" numerical model the errors should be independent and tend to the normal distribution with mean zero and variance $\sigma^2$. Error distribution in Fig. 2 has the features of Gaussian distribution, but in Table 1 for K=1, the means of dS and dT are neither zero nor equal. This already provides information about the model quality. Similar argumentation holds for the STD.

We cannot agree that approximation of a circular cluster is a good approximation, as the error distribution is skewed towards positive dT. In addition, if we assume circular clusters, then we loose relevant information about spatial and temporal quality of the model. Very roughly, if we presume that cluster k=4 is one cluster and cluster k=1,2,3 is the other cluster in Figure 5, then we do not get information that salinity is either over or underestimated while temperature is "correct" in the

southwestern Baltic. In addition, we lose information on the vertical and temporal structure of the errors.

5) We disagree that K-means clustering does not add any information on the structure of the data. The distribution of data is not Gaussian and to use arbitrary number of std for threshold is not justified. In this application we have shown that clustering provides meaningful information on the spatio-temporal distribution of the model errors. This is relevant for the interpretation of the model results and for the future improvement of the model.

6) We completely agree with this comment. We wanted to show the order of magnitude of comparison data that is needed for assessment of the model in present application. If the numerical model performance is stable, i.e. the data comes from the same distribution, then the location of the centroids does not change. But if the model quality "drifts away" from its initial quality, then the location of the centroids changes, which will be a warning signal for the uses of continuously run model like near-real-time ocean forecast model.

The sentence (line 274-275) "The rough estimate of the number of comparison points is about 100 000 for the current model, which shows relatively stable centroids and the stability of the model accuracy." is more correct.

The use of random location for initial clusters and performing multiple runs did not change the results.

---

## Author Comment (AC3)

**Answers to review's comments**

Dear reviewer, thank you for your valuable comments. Below we have addressed all of you concerns.

*My primary worries are as follows:*

*1. Clustering techniques are frequently used to evaluate atmospheric models, biogeochemical models, and so on. The variables in those models are multidimensional and, to an extent, "colossal." Typically, the output of an ocean circulation model is not regarded as a massive dataset. To persuade me to experiment with various clustering approaches based on machine learning, the interpretation of the clusters should be striking.*

In the elaboration of the K-means clustering method for assessment of the ocean general circulation model (GETM in particular case), we used two essential variables – salinity and temperature. These variables "integrate" temporal and spatial dynamics of the water basin that has been modelled. Usually temperature and salinity are measured simultaneously. To form error space of the model, the assumption is that different variables are measured simultaneously. Already now we had more than one million data pairs. In the interpretation of the error clusters, we limited ourselves to the main physical features of the Baltic Sea. It is known that the circulation models have problems in reproducing the highlighted dynamics "*poorly simulated thermocline (increasing vertical resolution), Baltic inflow problem (increasing bottom inflow), Danish strait problem (too close to open boundary), river temperature problem (no easy solution), SST problem (bulk formula)*". These features were clearly shown by the K-means clustering method, which shows the applicability of the method in assessment of the model quality.

In the assessment of atmospheric models, the set of simultaneously measured variables could be pressure, temperature, humidity, (wind speed), which forms 4-dimensional error space. Indeed, then the number of error quadruplets is much larger. In the marine biogeochemical models, essential variables are nitrate, phosphate and dissolved oxygen, which are usually measured simultaneously. These variables somehow "integrate" biology and chemistry of the model. In the coupled physical and biogeochemical models, it is natural to form 5-dimensional error space (temperature, salinity, nitrate, phosphate and dissolved oxygen) for the assessment of the model system, as biogeochemistry depends on the physics, also. For different models, geographical region and time period, the number of multidimensional error points could be very large.

*2. Prior impacts on clustering approaches, particularly hierarchical clustering methods, should be acknowledged. Without a doubt, comparing hierarchical clustering against centroid-based clustering is worthwhile.*

We tried to perform agglomerative clustering for the whole dataset. The outcome was that too much computer memory and computational time was needed. We performed agglomerative clustering for the surface layer data, only. There was no significant difference between the results of the K-means algorithm and hierarchical clustering algorithm (except computational resources) for the surface layer data. Thus, we consider that using K-means algorithm is computationally more feasible than using hierarchical clustering.

In cases of the K-means algorithm we have to select the number of clusters, while in case of hierarchical clustering the distance between clusters should be predefined (Hastie, T., Tibshirani, R., Friedman, J., 2009. The Elements of Statistical Learning. Data Mining, Inference, and Prediction. Springer, 745 pp.), which is not straightforward. In the latter case, for selection of the number of clusters, we had to plot distribution of the clusters in the error space and decide if the clusters have reasonable oceanographic meanings. Usually, dendrograms are used for the visualisation of the

results of hierarchical clustering, but in our case with the data number of O($10^5$), the visualisation of the results is not straightforward. Using of different algorithms for hierarchical clustering might be more justified, but comparison of different clustering algorithms is not the scope of this paper and requires separate study.

Without clustering our intuition for error clustering by setting thresholds… would be ambiguous. Therefore, we would need ML algorithm which would learn from data.

We add several sentences concerning hierarchical clustering methods. Main disadvantage of the hierarchical clustering methods is that they require more computational time and computer resources than K-means clustering algorithm.

*3. The cluster interpretation should emphasize the distinct outcomes using the Taylor and target diagrams. At the moment, I see no evidence of new information being obtained (my last comment).*

Some preliminary assumptions are needed to perform model validation using Taylor or target diagram. These methods require that existing measurements are somehow spatially or temporally grouped, e.g. we select all measurements over certain geographical area, calculate the statistics and present it as one point in the diagrams. This procedure will be applied for different regions or depth levels so that set of points will be displayed in the Taylor diagram. When applying this method, then the information about model performance within the spatial domain is lost. Using K-means clustering algorithm, the spatial and temporal (+seasonal) analysis of the errors is new (Fig. 5), for example. In comparison, Kärnä et al. (2021) (Kärnä, T., Ljungemyr, P., Falahat, S., Ringgaard, I., Axell, L., Korabel, V., Murawski, J., Maljutenko, I., Lindenthal, A., Jandt-Scheelke, S., Verjovkina, S., Lorkowski, I., Lagemaa, P., She, J., Tuomi, L., Nord, A., Huess, V., 2021. Nemo-Nordic 2.0: Operational marine forecast model for the Baltic Sea. Geoscientific Model Development 14(9), pp. 5731-5749. doi:10.5194/gmd-14-5731-2021) used conventional methods for validation of the NEMO-Nordic 2.0 circulation model. Their results on the spatial distribution of the model errors are presented on Fig. 8.

The second point how the Taylor and target diagrams differ from K-means clustering is that in case of Taylor diagram all variables are treated independently of the others. For instance, the statistics for salinity and temperature are calculated separately and form two points in Taylor and target diagram. In K-means a location of a single centroid is found, which represents model errors for interdependent salinity and temperature errors.

The K-means algorithm enables to assess the model performance over entire model domain and in time. For instance, Fig. 5. shows that at the eastern side of the Bothnian Sea, in certain case, the model overestimates temperature (cluster k=3), while being more correct in the open part of the Bothnian Sea. This information cannot be obtained if we use Taylor diagram, unless we calculate error statistics for the eastern coastal area of the Bothnian Sea and open Bothnian sea separately and present it in the Taylor diagram.

In a specific example, presented in Fig. 6a, we evaluate the model performance at the monitoring station BY15. K-means clustering approach, implemented on whole dataset, shows that below the halocline (depth>60-80m) the model underestimates salinity (errors belong to the cluster k=1) from 1966 to 1989. From 1990 to 2003 model has correct salinity, but temperature is slightly overestimated (errors belong to the cluster k=2). This information cannot be extracted from Taylor diagram, unless we calculate salinity and temperate errors for different depth intervals and different time periods, i.e. 1966-1989 and 1990-2003.

*4. The Baltic Sea is very special. The salinity is significantly lower than that of other marginal seas, and interaction with the open ocean is extremely limited, among other factors. I have my doubts about the method applied to the Baltic Sea being universally applicable; yet, this should be discussed.*

We agree that the Baltic Sea is different from the other marginal seas and the ocean. Still, we cannot follow the argument by reviewer that the method we propose could not be applied to the other seas or ocean. For instance, the same metrics is used for different seas (incl. the Baltic Sea) and for the ocean in CMEMS. If the reviewers concern is small salinity variability of world ocean or the other coastal seas compared to the Baltic Sea, then this should not impact clustering of the normalized salinity errors. To validate the proposed method for the other seas is a separate task.

*As a result, I recommend that the authors pursue two revision strategies for the paper. One possibility is to include more model data (sea level, mixed layer depth, currents, sea ice, and possibly heat fluxes and runoffs),*

In current stage of the elaboration of the K-means clustering algorithm for the assessment of the general ocean model quality (GETM in particular case), we form an error space using the set of **simultaneously** measured variables (temperature and salinity). Sea level, mixed layer depth, sea ice concentration and or thickness, heat fluxes are 3 dimensional (2D in space and time) fields. We use temperature and salinity, which are 4D (3D in space and time). Some of the suggested variables are not directly measurable (mixed layer depth, heat fluxes except solar radiation). It is rather difficult to obtain simultaneous measurements of sea level height, ice parameters unless we are limited to the coastal sea. River runoffs are completely different type of variable. Currents are measured at very selected locations and time and not necessarily simultaneously with temperature and/or salinity. We agree, that these variables could be included in the assessment of the models using K-means algorithm, but in future work.

*or to use multiple models (at least two, another one can be CMEMS results). This way, I can determine the method's reliability.*

We have used proposed K-means methods for the assessment of model quality in two papers, one is published and the other one is currently under revision. Indeed, both of the applications deal with the Baltic Sea.

In paper by Kõuts et al. (2021) (Kõuts, M., Maljutenko, I., Elken, J., Liu, Y., Hansson, M., Viktorsson, L., Raudsepp, U., 2021. Recent regime of persistent hypoxia in the baltic sea. Environmental Research Communications 3(7), 075004. doi: 10.1088/2515-7620/ac0cc4) we used proposed method for the assessment of coupled physical and biogeochemical model reanalyses data. The reanalyses data belong to the CMEMS multi-year product. The error pairs were formed for salinity and dissolved oxygen. In the paper by Kõuts et al. (2021), both the proposed method, common statistics and Taylor diagrams were used. The paper showed that more general picture of the model performance can be obtain with the proposed K-means method than with using Taylor diagram.

In paper by Raudsepp et al (under revision), we assessed the quality of the NEMO-Nordic 2.0 model performance (used in the CMEMS for near-real-time product) in reproducing surface temperature and salinity fields in comparison with ferry-box measurements along the ship track in the Baltic proper. The results showed that either model or ferry-box data cannot be trusted at the entrance area to the ports, especially in the southern Baltic Sea. This result could be intuitive, but in the study, we have shown it based on the data.

We provide reference to the paper by Kõuts et al. (2021) in revised manuscript.

*Another possibility is to incorporate additional clustering methods, such as agglomerative hierarchical clustering (bottom-up), divisive hierarchical clustering (top-down), or 'soft' K-means clustering (distribution-based) vs. rule-based methods (geographic areas, etc.). Clustering evaluation enables the acquisition of beneficial best-practices for clustering analysis. I believe that the work in these two areas does not require much time, and hence I recommend a major revision.*

We have done the experiments with agglomerative hierarchical clustering and with divisive hierarchical clustering. Main concern by applying these methods is that these methods are not so robust as the K-means clustering is. In addition, these methods need much more computational resources. We have used the other K-means algorithms as suggested by reviewer 1 and found no significant differences in the results. Rule-based algorithms have assumptions that follow prior knowledge of the rules, i.e. geographical regions, or use the other machine learning algorithm to define the rules.

In conclusion, the proposed method is simple and robust, feasible in terms of computer resources required and contains information for general assessment of the model quality as well as for task oriented posterior analysis. We address the concern of the reviewer in revised manuscript.

*Introduction:*

*P3, L40-L41: The rationale for using clustering methods is unclear. The shortcoming is that those papers did not include enough information in data? What is 4 dimensional information embedded? For instance, vertically, even if the vertically resolution in the observation is 1 cm, but you still bin to the resolution of 5m, don't you? You did not include more information than traditional methods. I feel that the problem of standard statistical metrics (Taylor and target diagram) is their inability to express clustered error statistics, such as error in climatology, seasonal, or diurnal signals. By the way, what are your criteria for defining 'the huge dataset'?*

We explain the advantages of clustering methods more clearly.

4 dimensional information is that error pairs can be mapped back to the (x,y,z,t) space for posterior analysis after clustering is done. We have interpolated the model data to the exact location and time of the measurements as they are in the database. Vertically, the 5-m bins and horizontally 25 km$^2$ grid are used for the analysis of the clustered errors.

We refer to the Fig. 5 in our study and Fig. 8. by Kärnä et al. (2021), as well as paper by Kõuts et al. (2021) to decide about the information that is obtained by traditional methods and the method proposed by us. Much more information can be obtained from the proposed method during postprocessing. Our aim was to show how the methods performs in obtaining general information on the model quality.

In the current context "the huge dataset" is dataset where the implementation of machine learning methods helps to extract and understand the information.

*P3, L49: It appears as though this 'K-means clustering algorithm' has fallen from the sky. This section should contain an introduction to conventional clustering algorithms. There is something missing at the start of L50.*

It has not fallen from sky, but has been adopted from clustering literature/text books (e.g. Hastie et al., 2009), where it has been straightforwardly introduced as robust and easily understandable to wider audience.

We rewrite this section adding the introduction to conventional clustering algorithms.

*P3, L60-64: This section should be in the 'discussion or perspective'. Why in the 'introduction'? Perhaps some previous efforts have already made used of it in an operational mode? Then they should be cited.*

We move this part to the discussion.

*P3, L68-70: This article discusses the results for the entire Baltic Sea. Other validation studies of GETM in the Baltic Sea, not just in the Gulf of Finland, should be cited.*

We also cite the other application of the GETM model in the Baltic Sea.

*Materials and Methods:*

*P4 Why this subsection 2.1 is in 'Methods'? It should be in the introduction part, and review of the Baltic Sea dynamics should be included, with a reference to the discussion in the subsequent section on 'adopting this method with caution' in other seas.*

The subsection 2.1 is moved to the introduction and we include short review of the Baltic Sea dynamics. Still, it is somehow unclear, why this method cannot be adopted to the other seas. In the future study, it is aimed to apply this method to the other European seas included in CMEMS.

*P6, L120, What is meant by a 'preliminary' check? That is, by examining Fig. 1a?*

Yes. We write it more clearly in the revised manuscript.

*P6, L127, 'This complicates data collection.' What does it mean? Perhaps you mean 'gathering of data during winter is very complicated'?*

Yes. We rewrite it as suggested.

*P6, L140, 'The squared Euclidean distance' is also coming from sky. Is that different clustering measures should be introduced and the reason to not choose nonEuclidean measures should be clearly stated.*

We include different measures in the description and justify why we use squared Euclidian distance. The square Euclidian distance is commonly used as the first choice of the measure of the distance, if not justified otherwise. We like to note, that we have normalized the salinity and temperature errors to make clustering independent on the data units. Thus, the clustering is performed to normalized errors, but the results are presented in original units. This also stated in the manuscript.

*Results:*

*P12, Figure5d, the dramatic change of clusters in recent years, e.g. big increase of K1, is it because of the smoothing you applied? BTW, add the meaning of pK in caption.*

The dramatic change of clusters is not due to smoothing. It is seen in Fig. 1b that number of measurements has increased at that time. This increase is mainly due to increased number of measurements in winter season. In winter, large volume inflows to the Baltic Sea occur. Model underestimates the salinity of these inflows and spreading of the water downstream in the Baltic Sea.

*P16, Section 3.4: Interpretation of the clusters. My concern 3 reflects the issue raised in this section. Almost all of the problems in this section can be well-defined using traditional methods and have generally recommended solutions, e.g. poorly simulated thermocline (increasing vertical resolution), Baltic inflow problem (increasing bottom inflow), Danish strait problem (too close to open boundary), river temperature problem (no easy solution), SST problem (bulk formula).*

We have provided evidence that our method provides information about model quality over entire spatial modelling domain and in time. It takes into account interdependent variables that describe the model performance in general. Also, we have shown that posterior analysis can provide information on model performance in specific area and time period. All this information cannot be obtained by using Taylor or targeted diagrams. On the other side proposed method is simple and robust. The interpretation of the results is straightforward concerning intuitive knowledge of the

modellers, but provides quantitative measures. Posterior analysis could fetch out different type of information on particular region of the model and time period of interest.

Computationally this method is feasible and can be applied on "colossal" datasets. We have provided postprocessing and interpretation of the result in different levels.

*Nothing novel! I would anticipate more new information if authors include more data than T and S. While one may argue that this is not critical, if not the primary need of GMD, it gives me, as a modeler, the feeling that this method is unnecessary.*

We present new method here not investigate the new findings from model. That's why we select model where we know main errors a-priori. By using K means single-handedly, we have identified all known errors without examining each region and depth layer separately (see. Maljutenko and Raudsepp 2014).

---

## Author Response (AR1)

**Answers to reviewer 1 comments**

We very much appreciate your comments on the unsupervised machine learning aspects of the model validation method. We are familiar with the book "The Elements of Statistical Learning. Data Mining, Inference, and Prediction" by Trevor Hastie, Robert Tibshirani, Jerome Friedman.

The model errors do not have obvious clusters unless there are obvious errors in the model. Our clustering strategy was to get meaningful spatio-temporal distribution of the model errors based on the distribution of data in the clusters. In addition, common statistics that have been used so far in the validation of the models can be calculated for each cluster. We wanted to keep the clustering procedure relatively simple and easy to implement for a wider audience who are not experts in the field of unsupervised machine learning. Another positive aspect of using K-means, instead of other algorithms, e.g., kernel K-means, was the relatively low computational time. Usually, in ocean model validation we deal with a huge dataset.

While doing research and preparing the manuscript, our test showed that by using random clusters and running the clustering until the clustering outcome remains consistent and using $O(10^{-4})$ criterion for the convergence, changes in the results were minor. We have added the following sentences in the manuscript:

"For practical reasons (Hastie et al., 2009), a regular pattern of initial centroids was chosen for this study (Fig. 2b), although we have run the algorithm with randomly spaced clusters."

"For practical reasons, the number of iterations was limited to 100, which ensured the convergence of the clustering algorithm."

Considering your concerns:

*1) The position of initial centroids for K-means is uniformly distributed and does not change in the paper. The K-means algorithm is heavily dependent on the choice of initial centroids. Hence, the algorithm may reach a local minima for the distance metric but it cannot be said that this minima is the global minima (or the "most optimal"). To overcome this dependence, the K-means algorithm is often run many times with randomly placed centroids. If the clustering outcome remains consistent, then it is seen as a relatively reliable outcome for the algorithm.*

We have made experiments with a) random selection of the initial centroids, b) random selection of the initial centroids in the range of min/max data rectangle and c) uniformly distributed initial centroids as described in the paper. In all cases, centroids converged almost to the same locations. Using uniformly distributed initial centroids was merely a suggestion from which to start and check if meaningful clusters in terms of numerical model under consideration occur. We have added the following sentence:

"For practical reasons (Hastie et al., 2009), a regular pattern of initial centroids was chosen for this study (Fig. 2b), although we have run the algorithm with randomly spaced clusters."

*2) There is no obvious "elbow" in the plot of the distance. Although this step can be thought of as suggestive, an elbow in the data should correspond to the number of clusters K = k such that the change in rate forms a jump in the rate function at K = k + 1. If the plot of minimum distance over K is connected across the histograms in figure 4, it has a smooth exponential decay which indicates that*

*there is no obvious cluster structure for K-means.*

We agree that there is no obvious elbow in the plot of the distance. But the distance does not have a smooth exponential decay. We calculated the first and second derivative of the distance as function of the number of clusters, which suggests using 2 or 4 clusters.

Indeed, we agree that there is no obvious cluster structure in our dataset. It was challenging for us to implement the clustering algorithm to the dataset without any obvious clusters. Our aim was to see if there are meaningful clusters in the context of the application.

3) *Halting the K-means process after 100 iterations does not "ensure convergence". Line 141 states that the number of iterations is limited for the numerical implementation of K-means to 100. This is not good numerical practice as it is not guaranteed that you will reach the local minima for the set of initial centroids in this number of iterations. Instead, it is numerically common to set a threshold for example, O(10–4 ) so that if the change in the distance metric is less than this for a given number of iterations then it is assumed to converge.*

We agree that halting the K-means process after 100 iterations does not "ensure convergence". All our tests showed that the K-means process converged before 100 iterations. Thus, we used 100 iterations as an indicative number of iterations. We have changed the text: "For practical reasons, the number of iterations was limited to 100, which ensured the convergence of the clustering algorithm."

4) *K-means requires the clusters in the space to be able to be divided by hyperplanes (lines in the case of R 2) and the distribution of error makes this difficult. Contrary to the statement made on line 173, error representation can actually provide some insight on the possible structure of clusters in the R2 error space. It is common in statistics to assume that the error of a model (like that of temperature and salinity in this case) is the sum of many independent errors, which by the central limit theorem tend to the normal distribution with mean zero and variance σ2. Of course there are exceptions to this in the case of dependence or outliers. Because we can reasonably assume a normal distribution for the errors of both temperature and salinity, we know if we plot these values on R2 that the majority of the points will be in the center with the number of observed points away from the origin decaying by the variance of their normal distribution. For an illustration, see a projection of the bivariate Gaussian distribution. This is also seen in figure 2.*

*The reason I mention the normality of the error distributions for temperature and salinity is because it is not possible to appropriately divide this type of resulting cluster structure by hyperplanes unless you increase the number of clusters K to some very large amount. In fact, the approximation of a circular cluster with a center can be seen in figure 3 as the number of clusters K increases. Given this, I would suggest that the authors look at other possibilities of clustering such as kernel K-means where the divisions can be made in a functional space. This may also give some reason as to why there is no obvious elbow in the distance vs. K graph.*

We agree that in case of an "ideal" numerical model the errors should be independent and tend to have normal distribution with mean zero and variance $\sigma^2$. Error distribution in Fig. 2 has the features of Gaussian distribution, but in Table 1 for K=1, the means of dS and dT are neither zero nor equal. This already provides information about the model quality. Similar argumentation holds for the STD.

We cannot agree that approximation of a circular cluster is a good approximation, as the error distribution is skewed towards positive dT. In addition, if we assume circular clusters, then we lose relevant information about the spatial and temporal quality of the model. Very roughly, if we presume that cluster k=4 is one cluster and cluster k=1,2,3 is the other cluster in Figure 5, then we do not get information on whether salinity is overestimated or underestimated while temperature is "correct" in the southwestern Baltic. In addition, we lose information on the vertical and temporal structure of the errors.

5*) K-means clustering does not add any information on the structure of the data. Many of the results in section 3.2 can be found without clustering. The spatial locations provided in figure 5 and discussed in section 3.2 where overand under- estimates of temperature and salinity occur in the model can be found by setting a threshold of over- and under- estimation, say 2 standard deviations away from the mean, for each measurement and calculating the proportion of points in the (lat, lon) space that fall above or below the set thresholds.*

We disagree that K-means clustering does not add any information on the structure of the data. The distribution of data is not Gaussian and using an arbitrary number of STD for threshold is not justified. In this application, we have shown that clustering provides meaningful information on the spatio-temporal distribution of the model errors. This is relevant for the interpretation of the model results and for the future improvement of the model.

6) *The process described in 3.3 just describes the continuation of the K-means algorithm. The K-means algorithm begins with a set of random centroids, assigns points to the centroids based on their proximity, recalculates the centroids based on the mean, and continues this way until a local minima of the inter-cluster distances is reached. If the algorithm is run on a uniformly selected subset of points coming from a fixed distribution (which is true in this case), the reassignment will continue until a local minima is obtained. If more points coming from the same distribution are added and the end centroids are used, the algorithm will continue from its final centroids until it reaches the exact clustering structure that is unique to the starting centroids, this is the case with figure 7 (a) (d). If less points are added, the structure remains constant (but the same since it corresponds to the same starting centroids), this is the case with figure 7 (b) (e) and (c) (f). This does not provide information on the stability of the clusters (line 274-275) which depends on changing the starting centroids and performing multiple runs of the K-means algorithm.*

We completely agree with this comment. We wanted to show the order of magnitude of comparison data that is needed for the assessment of the model in the present application. If the numerical model performance is stable, i.e., the data comes from the same distribution, then the location of the centroids does not change. But if the model quality "drifts away" from its initial quality, then the location of the centroids changes, which will be a warning signal for the uses of a continuously run model like the near-real-time ocean forecast model. Because of the last argument, we would like to keep the section in the manuscript.

The sentence (line 274-275) "The rough estimate of the number of comparison points is about 100 000 for the current model, which shows relatively stable centroids and the stability of the model accuracy." is more correct.

The use of a random location for initial clusters and performing multiple runs did not change the results.

*Additionally, there are some other minor issues with the article such as grammatical issues in the switching of "the" and "a" as in the first sentence of the abstract, present tense writing should be used when describing the work done for this article, and some spelling issues (e.g. line 50).*

The manuscript has gone through proofreading and the errors have been corrected.

*There is a lack of literature on clustering methods with Jain (2010) being the main reference.*

We have added the reference to Hastie et al. (2009) as the main reference: "Therefore, we suggest a new method based on the machine learning K-means clustering algorithm (Hastie et al., 2009; Jain, 2010) that takes advantage of a large set of available data and retains detailed spatial and temporal distribution of model errors that can be used for the posterior analysis of model accuracy."

Also, "The K-means clustering algorithm is a widely used algorithm in unsupervised machine learning (Hastie et al., 2009;Jain, 2010)."

*As I mentioned, I would suggest starting with the book by Friedman, Tibshirani, and Hastie for a foundational understanding of unsupervised learning algorithms.*

We have added references to the book by Hastie et al. (2009) where appropriate. We have tested several other algorithms and added the sentence to the Summary:

"Although the tests with the balanced iterative reducing and clustering using hierarchies (Zhang et al., 1996), the Gaussian mixture model and K-nearest neighbor algorithm (e.g. Hastie et al., 2009) were performed (results not shown), we have implemented the K-means algorithm because of its simplicity and robustness."

Basically, the same results were obtained with different algorithms.

**Answers to reviewer 2 comments**

Dear reviewer, thank you for your valuable comments. Below we have addressed all of your concerns.

*My primary worries are as follows:*

*1. Clustering techniques are frequently used to evaluate atmospheric models, biogeochemical models, and so on. The variables in those models are multidimensional and, to an extent, "colossal." Typically, the output of an ocean circulation model is not regarded as a massive dataset. To persuade me to experiment with various clustering approaches based on machine learning, the interpretation of the clusters should be striking.*

We agree that clustering has been used for the evaluation of the atmospheric, hydrological and biogeochemical model outputs, but, to our knowledge, not for the evaluation of model skills.

In the elaboration of the K-means clustering method for assessment of the ocean general circulation model (GETM in particular case) skills, we used two essential variables – salinity and temperature. These variables "integrate" temporal and spatial dynamics of the circulation in the water basin that has been modeled. Usually, temperature and salinity are measured simultaneously. To form the error space of the model, the assumption is that different variables are measured simultaneously. We already had more than one million data pairs. In the interpretation of the error clusters, we limited ourselves to the main physical features of the Baltic Sea. It is known that the circulation models have problems in reproducing the highlighted dynamics "*poorly simulated thermocline (increasing vertical resolution), Baltic inflow problem (increasing bottom inflow), Danish strait problem (too close to open boundary), river temperature problem (no easy solution), SST problem (bulk formula)*". These features were clearly shown by the K-means clustering method, which shows the applicability of the method in the assessment of model quality.

In the assessment of atmospheric models, the set of simultaneously measured variables could be pressure, temperature, humidity, (wind speed), which form a 4-dimensional error space. Indeed, then the number of error quadruplets is much larger. In marine biogeochemical models, essential variables are nitrate, phosphate and dissolved oxygen, which are usually measured simultaneously. These variables somehow "integrate" the biology and chemistry of the model. In the coupled physical and biogeochemical models, it is natural to form a 5-dimensional error space (temperature, salinity, nitrate, phosphate and dissolved oxygen) for the assessment of the model system, as biogeochemistry also depends on the physics. For different models, geographical region and time period, the number of multidimensional error points could be very large.

In the answer to the general comment 3, we show that by using K-means clustering we obtain distribution of the model errors in the southwestern Baltic Sea. Sometimes the model overestimates salinity (30%) and sometimes underestimates salinity (30%) there (Fig. 5b). We agree that intuitively this is what could be expected. If we include Fig 5e, then the data suggests that salinity could be underestimated in winter and overestimated in summer. This information could not be obtained with conventional methods without focused efforts. We would like to note that these results are obtained by performing the K-means algorithm for the entire Baltic Sea only once.

L230-232 "Vertical distribution of the error clusters confirms that the share of "good match" errors ranges between 0.5 and 0.9 of all data (Fig. 5e). In the surface layer, we have "overestimated salinity" and "underestimated salinity" in almost 50% of cases. In comparison with horizontal distribution of errors, a large part of these errors probably belong to the Danish straits (Fig. 5b)."

L245-L247 "Seasonally "overestimated salinity" has a higher share in summer, while "underestimated salinity" has a higher share in winter (Fig. 5d). Combining horizontal (Fig. 5b) and seasonal

distribution of errors (Fig. 5d), we could conclude that the salinity is overestimated in the Danish straits in summer and underestimated in winter."

L303-L304 "Model accuracy is relatively low in the Danish straits. The model has "underestimated salinity" in winter and "overestimated salinity" in summer (bias of 3.44 g kg$^{-1}$ and standard deviation of 1.59 g kg$^{-1}$) there."

In the answer to the general comment 3, we show that time-depth distribution of the model errors at BY15 can be obtained by performing the K-means clustering algorithm once for the entire Baltic Sea and doing posterior analysis of the model errors at a specific location. Obtaining similar information using conventional methods is not straightforward.

L258-268 "We extract error profiles from Gotland Deep station BY15, which is widely used for the validation of the physical and biogeochemical models of the Baltic Sea. In the upper layer of 60 m, the model has "good match" (Fig. 6a,b). There are isolated occasions of 10% in total when the model "overestimates temperature" in the seasonal thermocline (Fig. 6b). At the depth range 60-100 m, the share of model "underestimating salinity" increases. From a depth of 100 m, the proportion of the model that "underestimates salinity" gradually increases with depth. The Howmüller diagram shows that there are extended time periods when the model "underestimates salinity" (Fig. 6a). In the surface layer, the model has "good match", although model salinity starts to deviate from the measurements from 1995 onwards (Fig. 6c,e). At the bottom, the model reproduces temperature very well at the end of 1970s and beginning of 1980s, but as salinity is underestimated, the errors belong to the cluster of "underestimated salinity" (Fig. 6d,f). In general, the model has "good match" in the water column from 1991 to 2003 (Fig 6a,f). Dynamically, this corresponds to the end of the stagnation period and recovery of the bottom salinity and strengthening of the permanent halocline."

*2. Prior impacts on clustering approaches, particularly hierarchical clustering methods, should be acknowledged. Without a doubt, comparing hierarchical clustering against centroid-based clustering is worthwhile.*

We tried to perform agglomerative clustering for the whole dataset. This required too much computer memory and computational time. We performed agglomerative clustering only for the surface layer data. There was no difference between the results of the K-means algorithm and the hierarchical clustering algorithm (except computational resources) for the surface layer data. Thus, we consider that using the K-means algorithm is computationally more feasible than using hierarchical clustering.

With the K-means algorithm we have to select the number of clusters, while with hierarchical clustering the distance between clusters should be predefined (Hastie, T., Tibshirani, R., Friedman, J., 2009. The Elements of Statistical Learning. Data Mining, Inference, and Prediction. Springer, 745 pp.), which is not straightforward. In the latter case, in order to select the number of clusters, we had to plot distribution of the clusters in the error space and decide if the clusters have reasonable oceanographic meanings. Usually, dendrograms are used for the visualisation of the results of hierarchical clustering, but in our case, with the data number of O(10$^6$), the visualisation of the results is not straightforward. Using different algorithms for hierarchical clustering might be more justified, but a comparison of different clustering algorithms is not within the scope of this paper and requires a separate study.

We have repeated the exercise using Birch algorithms of hierarchical clustering. The results did not change. Th main disadvantage of hierarchical clustering methods is that they require more computational time and computer resources than the K-means clustering algorithm.

We added several sentences concerning hierarchical clustering methods.

L56-57 Introduction "Indeed, other clustering methods could be implemented, e.g., hierarchical clustering."

L336-338, Summary "Although the tests with the balanced iterative reducing and clustering using hierarchies (Zhang et al., 1996), the Gaussian mixture model and K-nearest neighbor algorithm (e.g. Hastie et al., 2009) were performed (results not shown), we have implemented the K-means algorithm because of its simplicity and robustness."

*3. The cluster interpretation should emphasize the distinct outcomes using the Taylor and target diagrams. At the moment, I see no evidence of new information being obtained (my last comment).*

Some preliminary assumptions are needed to perform model validation using the Taylor or target diagram. These methods require that existing measurements are somehow spatially or temporally grouped, e.g., we select all measurements over a certain geographical area, calculate the statistics and present it as one point in the diagrams. This procedure will be applied for different regions or depth levels so that the set of points will be displayed in the Taylor diagram. When applying this method, the information about model performance within the spatial domain is lost. Using the K-means clustering algorithm, the spatial and temporal (+seasonal) analysis of the errors is new (Fig. 5), for example. In comparison, Kärna et al. (2021) (Kärnä, T., Ljungemyr, P., Falahat, S., Ringgaard, I., Axell, L., Korabel, V., Murawski, J., Maljutenko, I., Lindenthal, A., Jandt-Scheelke, S., Verjovkina, S., Lorkowski, I., Lagemaa, P., She, J., Tuomi, L., Nord, A., Huess, V., 2021. Nemo-Nordic 2.0: Operational marine forecast model for the Baltic Sea. Geoscientific Model Development 14(9), pp. 5731-5749. doi:10.5194/gmd-14-5731-2021) used conventional methods for validation of the NEMO-Nordic 2.0 circulation model in the Baltic Sea. Their results on the spatial distribution of the model errors are presented on their Fig. 8, which could be compared with our Fig. 5b.

The second difference the Taylor and target diagrams have compared to K-means clustering is that in the case of the Taylor diagram, all variables are treated independently of the others. For instance, the statistics for salinity and temperature are calculated separately and form two points in the Taylor and target diagram. In K-means, a location of a single centroid is found, which represents model errors for interdependent salinity and temperature errors.

The K-means algorithm enables to assess the model performance over an entire model domain and in time. For instance, Fig. 5. shows that at the eastern side of the Bothnian Sea, in a certain case, the model overestimates temperature (cluster k=3) while being more correct in the open part of the Bothnian Sea. This information cannot be obtained using the Taylor diagram, unless we calculate error statistics for the eastern coastal area of the Bothnian Sea and open Bothnian Sea separately and present it in the Taylor diagram.

As an example, we calculated the salinity and temperature bias for the southwestern Baltic Sea (see Fig An1 for the area). The bias for salinity was dS: 0.23 g/kg and for temperature dT: 0.66 ℃. From this, we could conclude that the model performs quite well in the southwestern Baltic Sea. Using K-means clustering, we showed that in up to 30% of the cases, the model has underestimated salinity (k=1; dS=-1.96 g/kg) there, and in up to the other 30% of cases, the model has overestimated salinity (k=2; dS=3.44) there (Fig. 5b, Table 1).

[Figure]

Figure An1. The green colour represents the area of the Baltic Sea for which the model temperature and salinity bias was calculated. The calculation was performed for all data points that fell into the area and time period of the model run (1966-2006).

In a specific example, presented in Fig. 6a, we evaluate the model performance at the monitoring station BY15. The K-means clustering approach, implemented on the whole dataset, shows that below the halocline (depth>60-80m) the model underestimates salinity (errors belong to the cluster k=1) from 1966 to 1989. From 1990 to 2003, the model has correct salinity but temperature is slightly overestimated (errors belong to the cluster k=2). This information cannot be extracted from the Taylor diagram, unless we calculate salinity and temperate errors for different depth intervals and different time periods, i.e., 1966-1989 and 1990-2003.

As an example, we calculated vertical profiles of temperature and salinity bias at the monitoring station BY15 (Fig. An2). This could be compared with the model performance there, as shown using the K-means clustering algorithm (Fig. 6a,b). The information about the model skills at BY15, i.e., variations of the model errors in time-depth (Fig. 6a), could not be easily obtained using conventional methods.

[Figure]

Figure An2. Vertical profiles of temperature and salinity bias of the model at BY15 calculated for the time period of the model run (1966-2006).

*4. The Baltic Sea is very special. The salinity is significantly lower than that of other marginal seas, and interaction with the open ocean is extremely limited, among other factors. I have my doubts about the method applied to the Baltic Sea being universally applicable; yet, this should be discussed.*

We agree that the Baltic Sea is different from other marginal seas and the ocean. Still, we cannot follow the argument by the reviewer that the method we propose could not be applied to other seas or the ocean. For instance, the same conventional metrics are used for different seas (incl. the Baltic Sea) and for the ocean in CMEMS. If the reviewer is concerned with regards to the small salinity variability between the world ocean or other coastal seas and the Baltic Sea, then this should not impact clustering of the normalized salinity errors. To validate the proposed method for the other seas is a separate task.

*As a result, I recommend that the authors pursue two revision strategies for the paper. One possibility is to include more model data (sea level, mixed layer depth, currents, sea ice, and possibly heat fluxes and runoffs),*

In the current stage of the elaboration of the K-means clustering algorithm for the assessment of the general ocean model quality (GETM in particular case), we form an error space using the set of **simultaneously** measured variables (temperature and salinity). Sea level, mixed layer depth, sea ice concentration and or thickness and heat fluxes are 3-dimensional (2D in space and time) fields. We use temperature and salinity, which are 4D (3D in space and time). Some of the suggested variables are not directly measurable (mixed layer depth, heat fluxes with the exception of solar radiation). It is rather difficult to obtain simultaneous measurements of sea level height and ice parameters unless we are limited to the coastal sea. River runoffs are a completely different type of variable. Currents are measured at very selected locations and times and not necessarily simultaneously with temperature and/or salinity. We agree that these variables could be included in the assessment of the models using K-means algorithm but in future work.

*or to use multiple models (at least two, another one can be CMEMS results). This way, I can determine the method's reliability.*

We have used proposed K-means methods for the assessment of model quality in two papers. One is published and the other one is revised and sent to the reviewers for the second round. Indeed, both of the applications deal with the Baltic Sea.

In the paper by Kõuts et al. (2021) (Kõuts, M., Maljutenko, I., Elken, J., Liu, Y., Hansson, M., Viktorsson, L., Raudsepp, U., 2021. Recent regime of persistent hypoxia in the baltic sea. Environmental Research Communications 3(7), 075004. doi: 10.1088/2515-7620/ac0cc4), we used the proposed method for the assessment of coupled physical and biogeochemical model reanalyses data. The reanalyses data belong to the CMEMS multi-year product. The error pairs were formed for salinity and dissolved oxygen. In the paper by Kõuts et al. (2021), both proposed methods (common statistics and Taylor diagrams) were used. The paper showed that a more general picture of the model performance can be obtained with the proposed K-means method than by using the Taylor diagram.

In the paper by Raudsepp et al (under the second round of review), we assessed the quality of the NEMO-Nordic 2.0 model performance (used in the CMEMS for near-real-time product) in reproducing surface temperature and salinity fields in comparison with ferry-box measurements along the ship track in the Baltic proper. The results showed that neither model nor ferry-box data can be trusted at the entrance area to the ports, especially in the southern Baltic Sea. This result could be intuitive, but in the study we have shown it based on the data.

We provided reference to the paper by Kõuts et al. (2021) in the revised manuscript and added a sentence.

L371-372 "An application of the method for the assessment of a coupled physical and biogeochemical model of the Baltic Sea is presented in Kõuts et al. (2021)."

*Another possibility is to incorporate additional clustering methods, such as agglomerative hierarchical clustering (bottom-up), divisive hierarchical clustering (top-down), or 'soft' K-means clustering (distribution-based) vs. rule-based methods (geographic areas, etc.). Clustering evaluation enables the acquisition of beneficial best-practices for clustering analysis. I believe that the work in these two areas does not require much time, and hence I recommend a major revision.*

We have done the experiments with agglomerative hierarchical clustering and with divisive hierarchical clustering. Our main concern in applying these methods is that these methods are not as robust as the K-means clustering. In addition, these methods need much more computational resources. We used the algorithms suggested by reviewer 1 and found no significant differences in the results. Rule-based algorithms have assumptions that follow prior knowledge of the rules, i.e., geographical regions, or use the other machine learning algorithm to define the rules.

In conclusion, the proposed method is simple and robust, feasible in terms of computer resources required and contains information for general assessment of the model quality as well as for task oriented posterior analysis.

L335-338 "In general, any other partitional clustering algorithm can be used instead of K-means for the clustering of multivariate model errors. Although the tests with the balanced iterative reducing and clustering using hierarchies (Zhang et al., 1996), the Gaussian mixture model and K-nearest neighbor algorithm (e.g. Hastie et al., 2009) were performed (results not shown), we have implemented the K-means algorithm because of its simplicity and robustness."

*Introduction:*

*P3, L40-L41: The rationale for using clustering methods is unclear. The shortcoming is that those papers did not include enough information in data? What is 4 dimensional information embedded? For instance, vertically, even if the vertically resolution in the observation is 1 cm, but you still bin to the resolution of 5m, don't you? You did not include more information than traditional methods. I feel that the problem of standard statistical metrics (Taylor and target diagram) is their inability to express clustered error statistics, such as error in climatology, seasonal, or diurnal signals. By the way, what are your criteria for defining 'the huge dataset'?*

We have rewritten and restructured the Introduction to be more clear.

We have deleted sentence with the term "4-dimensional information embedded".

We have interpolated the model data to the exact location and time of the measurements as they are in the database. 4-dimensional information means that error pairs can be mapped back to the (x,y,z,t) space for posterior analysis after clustering is done. The model bias is assigned to each location and time of the error pair according to the coordinates of the centroid to where the error pair belong. Vertically 5 m bins and horizontally a 25 km$^2$ grid are used for the analysis of the clustered errors. If the vertical resolution of the measurements is 1 cm, then the same resolution is kept for the errors and model validation.

See answers to the general comments 1 and 3 where we explain in more detail the advantages of the clustering algorithm compared to conventional methods. We refer to Fig. 5 in our study and Fig. 8. by Kärnä et al. (2021) as well as the paper by Kõuts et al. (2021) to come to a conclusion regarding the information obtained by traditional methods and the method proposed by us. Much more information can be obtained from the proposed method during postprocessing. Our aim was to show

how the method performs in obtaining detailed information on the model quality over the whole model domain and time span of the model.

In the current context, "the huge dataset" is a dataset where the implementation of machine learning methods helps to extract and understand the information. We do not use term "huge dataset" in the revised manuscript.

*P3, L49: It appears as though this 'K-means clustering algorithm' has fallen from the sky. This section should contain an introduction to conventional clustering algorithms. There is something missing at the start of L50.*

It has not fallen from the sky. It has been adopted from clustering literature/text books (e.g. Hastie et al., 2009), where it has been straightforwardly introduced as robust and easily understandable to a wider audience.

L51-57 "Ideally, researchers like to know the model accuracy for the whole model domain and time period considered. Therefore, we suggest a new method based on the machine learning K-means clustering algorithm (Hastie et al., 2009; Jain, 2010) that takes advantage of a large set of available data and retains detailed spatial and temporal distribution of model errors that can be used for the posterior analysis of model accuracy. This method belongs to the category of multivariate comparison. According to Hastie et al. (2009): "The K-means algorithm is one of the most popular iterative descent clustering methods. It is intended for situations in which all variables are of the quantitative type". Indeed, other clustering methods could be implemented, e.g., hierarchical clustering."

*P3, L60-64: This section should be in the 'discussion or perspective'. Why in the 'introduction'? Perhaps some previous efforts have already made used of it in an operational mode? Then they should be cited.*

To our knowledge, there have not been previous efforts in an operational mode. We have rewritten this section to be suitable for the Introduction as follows:

L70-75 "Additionally, we implement the learning-predicting sequence in the form of clustering stability tests. The learning period consists of the model run for a certain period and error clustering. The learning period is for determining the number of clusters and the coordinates of the centroids. Based on the error clustering of the learning period, we can presume that a similar error distribution is valid for the forward model simulation results. During the predicting period, new available errors are added to the clusters. The coordinates of the centroids and other metrics are updated. In the operational applications, the value of this process lies in the fact that the exploitation of model simulation results can start before new validation is completed."

*P3, L68-70: This article discusses the results for the entire Baltic Sea. Other validation studies of GETM in the Baltic Sea, not just in the Gulf of Finland, should be cited.*

We have deleted these sentences and this paragraph has been rewritten as follows:

L76-78 "We apply proposed K-means clustering methods for the assessment of the model quality of the General Estuarine Transport Model (GETM; Burchard and Bolding, 2002) of the Baltic Sea. In this particular application, the model is used for the hindcast simulation of the general circulation of the Baltic Sea in 1966–2006 (Maljutenko and Raudsepp, 2019)."

*Materials and Methods:*

*P4 Why this subsection 2.1 is in 'Methods'? It should be in the introduction part, and review of the Baltic Sea dynamics should be included, with a reference to the discussion in the subsequent section on 'adopting this method with caution' in other seas.*

The subsection 2.1 is moved to the introduction, L78-105, and we include a short review of the dynamics of the Baltic Sea. Still, it is somewhat unclear why this method cannot be adopted to other seas. In a future study, it is aimed to apply this method to other European seas included in CMEMS.

*P6, L120, What is meant by a 'preliminary' check? That is, by examining Fig. 1a?*

Yes. We have removed the sentence.

*P6, L127, 'This complicates data collection.' What does it mean? Perhaps you mean 'gathering of data during winter is very complicated'?*

Yes. We rewrote it as suggested.

L137-138 "Gathering data during winter is very complicated due to seasonal ice coverage of the Baltic Sea (Raudsepp et al., 2020)."

*P6, L140, 'The squared Euclidean distance' is also coming from sky. Is that different clustering measures should be introduced and the reason to not choose nonEuclidean measures should be clearly stated.*

The square Euclidian distance is commonly used as the first choice of the measure of the distance, if not justified otherwise. We would like to note that we have normalized the salinity and temperature errors to make clustering independent of the data units. Thus, the clustering is performed on normalized errors, but the results are presented in original units. This is also stated in the manuscript. For clarification, we have added a sentence on why we use the squared Euclidean distance:

L151-154 "The squared Euclidean distance was used as the measure of the distance between data points and the centroid coordinates of the cluster. The squared Euclidean distance measured from the cluster centroid is the most commonly used partitioning criterion for continuous data (e.g., Kononenko and Kukar, 2007; Hastie et al., 2009)."

Kononenko, I., Kukar, M., 2007. Machine Learning and Data Mining. Elsevier. 454 pp.

Citation from Hastie et al., (2009) "The K-means algorithm is one of the most popular iterative descent clustering methods. It is intended for situations in which all variables are of the quantitative type, and squared Euclidean distance $d(x_i, x_i') = \sum_{j=1}^{p}(x_{ij} - x_{i'j})^2 = ||x_i - x_i'||^2$ is chosen as the dissimilarity measure."

*Results:*

*P12, Figure5d, the dramatic change of clusters in recent years, e.g. big increase of K1, is it because of the smoothing you applied? BTW, add the meaning of pK in caption.*

The dramatic change of clusters is not due to smoothing. It can be seen in Fig. 1b that number of measurements has increased at that time. This increase is mainly caused by an increase in the number of measurements in the winter season. In winter, large volume inflows to the Baltic Sea occur. The model underestimates the salinity of these inflows and spreading of the water downstream in the Baltic Sea, which results in an increase of the share of cluster k=1. We have added the meaning of pK in the caption:

"The share, p(k) represents the share of the error points belonging to the cluster k, is calculated as explained in Section 2.4."

*P16, Section 3.4: Interpretation of the clusters. My concern 3 reflects the issue raised in this section. Almost all of the problems in this section can be well-defined using traditional methods and have*

*generally recommended solutions, e.g. poorly simulated thermocline (increasing vertical resolution), Baltic inflow problem (increasing bottom inflow), Danish strait problem (too close to open boundary), river temperature problem (no easy solution), SST problem (bulk formula).*

In the answers to the general comments 1 and 3, we explain the advantages of the clustering algorithm compared to the conventional methods in more detail."

We have provided evidence that our method provides information about model quality over the entire spatial modeling domain and in time. It takes into account interdependent variables (temperature and salinity) that describe the circulation model performance in general. Also, we have shown that posterior analysis can provide information on model performance in a specific area and time period. All this information could probably be obtained using Taylor or targeted diagrams with considerable effort. Meanwhile, the proposed method is simple and robust. The interpretation of the results is straightforward concerning intuitive knowledge of the modelers, but it provides quantitative measures. Posterior analysis could fetch different types of information on a particular region of the model and a time period of interest. In this paper, we showed a fraction of posterior analysis of the clustered errors.

Computationally, this method is feasible and can be applied on "colossal" datasets. We have provided postprocessing and interpretation of the results in different levels.

*Nothing novel! I would anticipate more new information if authors include more data than T and S. While one may argue that this is not critical, if not the primary need of GMD, it gives me, as a modeler, the feeling that this method is unnecessary.*

We present the new method here. We do not investigate the new findings from the model. That is why we selected a model where we know the main errors *a priori*. By using K means single-handedly, we have identified all known errors without examining each region and depth layer separately (see. Maljutenko and Raudsepp 2014). Any posterior analysis at any region or time period under interest is possible.